# In silico design of a multi-epitope vaccine against *Cryptosporidium parvum* using structural and immunoinformatics approaches

Guneswar Sethi[1], Avinash Kant Lakra[2], Kirti Nirmal[3], Jeong Ho Hwang[4]*

1 Center for Large Animals Convergence Research, Korea Institute of Toxicology, Jeongeup-si, Jeollabuk-do, Korea, 2 Translational Health Science and Technology Institute, Faridabad, Haryana, India, 3 Department of Microbiology, University College of Medical Sciences, Delhi, India, 4 Division of Advanced Predictive Research, Center for Bio-Signal Research, Korea Institute of Toxicology, Daejeon, Republic of Korea

* jeongho.hwang@kitox.re.kr

## Abstract

### Background

*Cryptosporidium parvum* is a waterborne protozoan parasite responsible for diarrheal illness in humans and animals. The lack of effective vaccines and the emergence of antimicrobial resistance underscore the urgent need for novel prophylactic strategies.

### Methods

A structure-based immunoinformatics approach was used to design a multi-epitope subunit vaccine (MESV) targeting immunogenic regions of *C. parvum*. Three proteins, Cp15, Cp23, and CpP2, were selected based on antigenicity, non-allergenicity, non-homology with host proteins, and absence of transmembrane domains. B-cell, CD4$^+$, CD8$^+$, and IFN-γ-inducing epitopes were identified and screened for high antigenicity, non-allergenicity, and non-toxicity. To enhance immune recognition, the lipoprotein LprA, a TLR2 agonist, was fused at the N-terminus using an EAAAK linker, and a PADRE sequence was added to improve helper T-cell responses. Linkers were applied to ensure proper epitope separation and processing. Population coverage was analyzed to evaluate the distribution of HLA-restricted epitopes across global populations. Structural modeling and flexibility analysis (CABS-flex) were performed to assess construct stability. Interactions with TLR2 and TLR4 were examined *via* molecular docking and 100-ns molecular dynamics (MD) simulations, with MM-GBSA used to estimate binding free energies. Immune simulations predicted host immune responses, while codon optimization, *in silico* cloning, and mRNA secondary structure prediction assessed expression and transcript stability.

**Data availability statement:** All relevant data are within the manuscript and its Supporting information files.

**Funding:** This research was supported by the National Research Council of Science & Technology (NST) (Grant Number: CRC21022) and the Ministry of Science and ICT (Project Number: 2710008770; KK-2513-01). The funders had no role in study design, data collection and analysis, decision to publish, or preparation of the manuscript.

**Competing interests:** The authors declare that they have no competing interests.

## Results

The MESV showed strong binding to TLR2 (−1328.4 kcal/mol) and TLR4 (−1133.3 kcal/mol), with MD simulations confirming stable complexes. Immune simulations indicated robust antibody production, T-cell activation, cytokine release, and dendritic cell recruitment. The vaccine demonstrated global HLA population coverage of 95.92%, with favorable expression and mRNA folding profiles.

## Conclusion

The MESV construct demonstrated strong immunogenicity, structural stability, and broad population coverage, underscoring its potential as a promising vaccine candidate against *C. parvum*. Furthermore, experimental validation is warranted to confirm its safety and efficacy.

## 1. Introduction

*Cryptosporidium parvum* is a protozoan parasite that causes cryptosporidiosis, a gastrointestinal disease characterized by acute diarrhea in both humans and animals [1]. The infection is particularly severe in immunocompromised individuals, such as those with HIV/AIDS, and in neonates, where it can lead to life-threatening dehydration and malnutrition [2–4]. In livestock, particularly neonatal calves, lambs, and goat kids, it causes significant economic losses due to high morbidity, mortality, and reduced growth performance [5]. Cryptosporidiosis is primarily transmitted through the fecal-oral route *via* the ingestion of oocysts from infected hosts. Contaminated water and food are common sources of infection [6]. Once ingested, oocysts release invasive sporozoites that attach to and invade the epithelial lining of the gastrointestinal tract. These sporozoites are enveloped in an actin-rich membrane that shields them from host immune defenses. The parasite completes its complex life cycle within the host, alternating between asexual and sexual replication, ultimately generating two forms of oocysts. Thick-walled oocysts are shed through feces and play a key role in spreading infection *via* the environment, whereas thin-walled oocysts remain inside the host and are responsible for self-infection, contributing to the chronic nature of the disease [6,7].

Cryptosporidium comprises over 44 species and 120 genotypes, with *C. hominis* and *C. parvum* being most commonly associated with human infections [8,9]. Beyond its impact on human health, Cryptosporidiosis is particularly common in cattle, with prevalence estimates ranging from 11.7% to 78%, especially in pre-weaned calves [10–13]. In humans, prevalence differs across regions, reported at 14.1% in high-income countries and up to 31.5% in low-income regions [14,15]. The Global Enteric Multicenter Study (GEMS) underscored the severity of the disease, estimating nearly 202,000 child deaths annually in sub-Saharan Africa and South Asia [16,17]. In the United States, outbreaks recorded between 2009 and 2017 were largely linked to contact with infected cattle and childcare facilities [18]. Moreover, from 2010 to 2020, *C. parvum* was responsible for over 96% of foodborne morbidity cases, highlighting its significance as both a public health challenge and an economic burden [19].

At present, management of cryptosporidiosis primarily involves supportive measures such as fluid and electrolyte replacement, together with the antiparasitic agent nitazoxanide. Although nitazoxanide remains the only FDA-approved therapy, its effectiveness is limited, showing only partial benefit in immunocompetent patients and little to no activity in immunocompromised individuals, including those with HIV/AIDS [20]. Other agents, such as paromomycin, azithromycin, and clofazimine, have been tested as potential alternatives, but their outcomes have been inconsistent and often inadequate [21]. Furthermore, nitazoxanide is limited by adverse effects such as gastrointestinal disturbances and headache, reducing tolerability and adherence [22]. Despite its global impact, there is currently no licensed vaccine available for the prevention of cryptosporidiosis in humans. These therapeutic limitations underscore the urgent need for preventive strategies, with vaccines representing the most promising approach. Given the mucosal nature of Cryptosporidium infection, an ideal vaccine should stimulate both mucosal and systemic immune responses. Although live attenuated vaccines can elicit robust immunity, their use is restricted due to safety concerns, especially in immunosuppressed individuals [23]. Consequently, subunit and epitope-based vaccines derived from immunodominant antigens are considered safer and more targeted. These approaches have gained attention because of their ability to elicit strong, specific immune responses while minimizing adverse effects [24,25].

However, vaccine development for protozoan parasites such as *Cryptosporidium* remains challenging due to their complex life cycles, antigenic variation, and ability to evade host immune responses. While several approaches have shown promise, no subunit or epitope-based vaccine has yet progressed to licensed formulations, underscoring the need for continued refinement and innovation. In response to these challenges, researchers have explored epitope-based vaccine designs specifically targeting *C. parvum*. Alvaro et al. have highlighted the immunodominance of the CpP2 antigen, suggesting its potential as a promising vaccine candidate [26]. Experimental studies have shown that Cp15 and Cp23 surface antigens induce strong humoral and mucosal immune responses, with Cp15-based DNA vaccination providing passive protection in neonatal animals [27,28]. Additionally, three novel vaccine candidates, Cp15, profilin, and Cryptosporidium apyrase, were identified by Manque et al. using a reverse vaccinology approach, leveraging genomic data from *C. hominis* and *C. parvum* to select antigens capable of eliciting strong immune responses [29]. Complementing these findings, Dhal et al. developed a multi-epitope based vaccine (MEV) by selecting two signal peptides and five hypothetical proteins from *C. parvum*, demonstrating the potential of subunit vaccine strategies [30]. Recently, *in silico* studies have proposed a combination of multiple epitopes from sporozoite surface proteins to design vaccines capable of eliciting cellular and humoral immunity.

Building on these insights and addressing the gaps in previous approaches, we employed a comprehensive immunoinformatics approach to design a novel MESV targeting *C. parvum*. B and T-cell epitopes were predicted from Cp15, Cp23, and CpP2 antigens and rigorously assessed for antigenicity, allergenicity, and physicochemical properties. We aimed to construct a vaccine capable of inducing strong cell-mediated, humoral, and innate immune responses while minimizing the risks associated with live or whole-parasite vaccines. This strategy provides a promising foundation for developing safe and effective vaccines against cryptosporidiosis.

## 2. Materials and methods

### 2.1. Sequence retrieval

The protein sequences of *C. parvum* Iowa II strain were retrieved in FASTA format from the UniProt database [31]. Three proteins, Cp15 (UniProt ID: Q23728), Cp23 (UniProt ID: Q8ITU5), and CpP2 (UniProt ID: Q9U553), were selected based on their immunogenicity, lack of allergenicity, and absence of transmembrane helices. The antigenicity of the proteins was assessed using VaxiJen v.2.0, with the threshold set to 0.5 [32]. The AllerCatPro 2.0 online tool was used to predict allergenicity based on structural and sequence similarity to known allergens [33]. To reduce the potential for inducing autoimmune reactions, the selected proteins were analyzed against the human proteome using BLASTp with default settings [34]. Additionally, the presence of transmembrane helices in the protein sequences was predicted using TMHMM 2.0 [35].

An overview of the MESV construction strategy is illustrated in Fig 1, and S1 Table details all databases, tools, and web servers used in the study.

## 2.2. Screening of potential epitopes

B-cell epitopes were predicted using ABCpred [36] and BepiPred-3.0 [37] servers. ABCpred applies machine learning based on non-repetitive epitopes from the Bcipep database and non-epitopes from Swiss-Prot, with a threshold of 0.85 and an accuracy of 65.93%. BepiPred-3.0 uses language model embeddings to improve prediction, with a default window length of 16 for better precision. HTL epitopes were predicted using the IEDB MHC II binding tool with the NetMHCIIpan 4.1 EL algorithm, which is known for its reliable performance [38,39]. The prediction was performed using an HLA allele panel covering 99% of the global population (S2 Table), prioritizing peptides with the lowest binding percentile ranks. We used IFNepitope [40] and IL4Pred servers [41] to assess immune activation to predict IFN-γ and interleukin (IL)-4 induction. These tools helped identify epitopes that are likely to trigger a strong immune response, which is crucial for vaccine development. CTL epitopes were predicted using the IEDB MHC-I server with the NETMHCpan 4.1 EL method and a representative panel of HLA alleles [42]. The selected epitopes were subsequently analyzed for their antigenic potential, allergenic properties, and toxicity using VaxiJen v.2.0, AllerCatPro 2.0 [33], and ToxinPred [43], respectively.

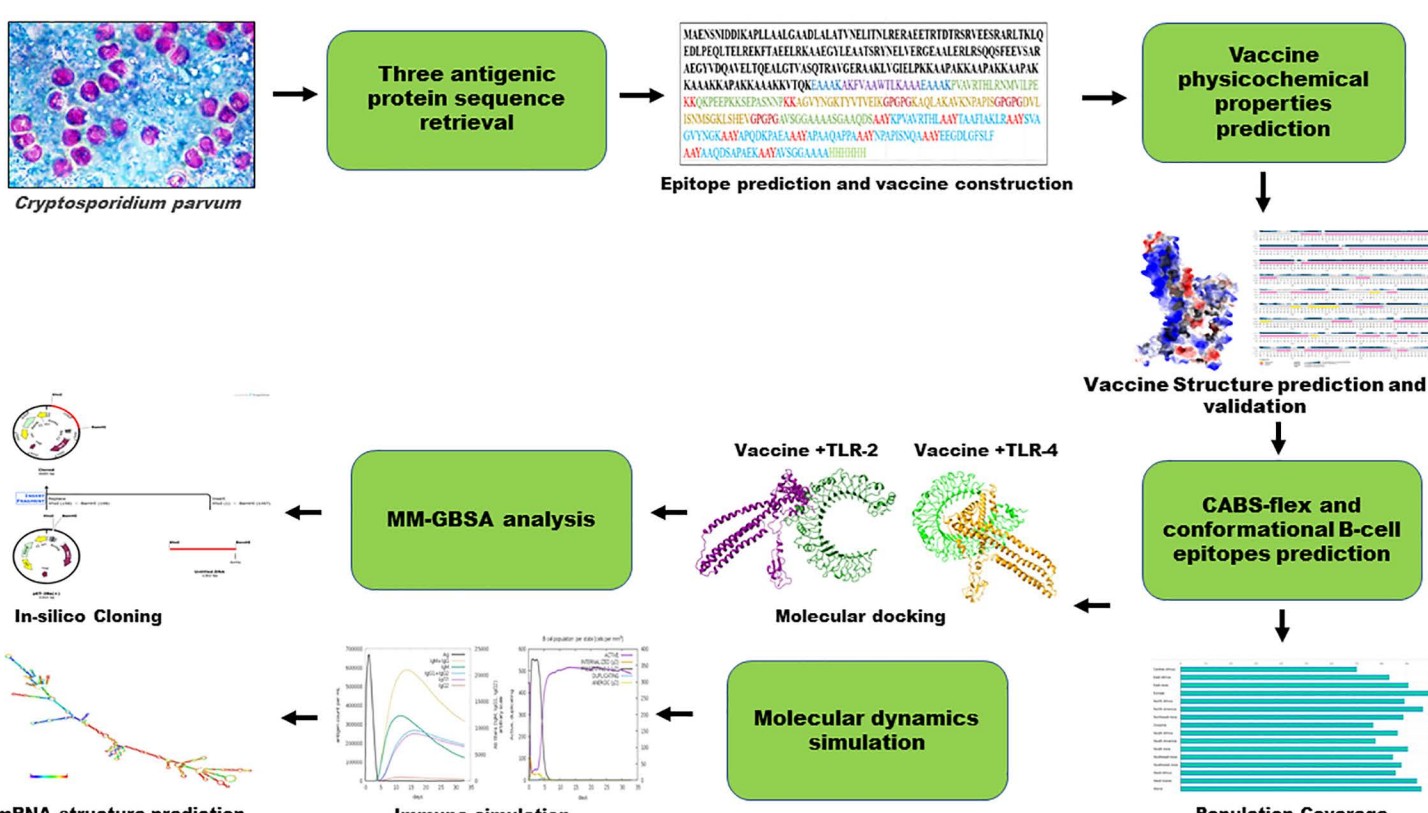

**Fig 1. Outline of the computational pipeline used for designing a multi-epitope vaccine against *Cryptosporidium parvum*.**

## 2.3. Vaccine construction and physicochemical properties

The MESV was designed by assembling carefully selected B-cell, CTL, and HTL epitopes, connected through appropriate linkers to support structural flexibility and functional independence. The selected epitopes demonstrated favorable properties, including strong antigenicity, immunogenicity, non-allergenicity, and non-toxicity. The lipoprotein LprA, a TLR2 agonist composed of 244 amino acids, was fused at the N-terminus *via* an EAAAK linker to boost immune recognition and enhance vaccine efficacy [44]. For optimal processing, CTL, HTL, and B-cell epitopes were joined using AAY, GPGPG, and KK linkers, respectively. These linkers were employed to preserve the epitope-specific activity, improve solubility, and promote proper folding of the vaccine construct. Additionally, the PADRE sequence and 6x His-tag were included to aid in protein expression and purification. The physicochemical properties of the MESV were analyzed using ProtParam [45]. Antigenicity was predicted using VaxiJen v.2.0, and allergenicity was assessed using AllerTop v.2.0. An additional antigenicity evaluation was performed using ANTIGENPro [42], and protein solubility was analyzed using the Protein-Sol server and SOLpro tool [46].

## 2.4. Structural analysis of vaccine

PSIPRED v4.0 [44] and GOR IV [45] servers were employed to analyze the secondary structure of the vaccine construct, both of which leverage neural network algorithms and information theory-based models. Tertiary structure modeling was performed using trRosetta, a deep learning-based tool built on the Rosetta framework, known for its accuracy in predicting complex protein conformations [47]. The model was refined using the GalaxyRefine2 web server to improve quality [48]. The quality of the final model was validated using a Ramachandran plot generated using PROCHECK, and further assessed using ProSA-web [49,50]. UCSF Chimera version 1.17.1 was employed to visualize the 3D structure and analyze the structural features of the vaccine [51].

## 2.5. CABS-flex analysis

CABS-flex 2.0 was used to perform coarse-grained simulations and analyze the structural flexibility of the peptide vaccine. The analysis was conducted over 50 cycles with 8335 RNG seeds. The analysis employed a temperature range of 1.40, 1.0 for the global side chain restraint, and 1.0 for the global C-alpha restraint. The AGGRESCAN 3D server v2.0 was used as a preliminary step in this investigation to identify aggregation-prone regions in the vaccine sequence [52,53].

## 2.6. Conformational prediction of B-cell epitopes

The ElliPro server was used to predict B-cell epitopes from the refined 3D vaccine structure [54]. It assigns an average Protrusion Index to each epitope, with a threshold of 0.9 for residues within 90% of the ellipsoid of the protein. The accuracy of ElliPro's structure-based epitope prediction was supported by an area under the curve value of 0.732, highlighting its reliability.

## 2.7. Population coverage and conservancy analysis

Considering the variability of HLA alleles among different populations, it is essential to design a vaccine that ensures broad immunological coverage. The IEDB population coverage tool [55], operated with default parameters, was used to evaluate the global reach of the predicted CTL and HTL epitopes by analyzing their interaction with prevalent HLA genotypes across regions. To assess epitope conservation, protein sequences from various pathogenic strains were aligned using MEGA version 7.0 with default parameters [56]. Conserved regions within the predicted epitopes were visualized using WebLogo v3.7.9 [57], which generated sequence logos illustrating amino acid conservation and frequency patterns. WebLogo was operated using the default settings.

 

## 2.8. Molecular docking study

To evaluate binding affinity, molecular docking between the vaccine and human TLRs was performed. Interactions between TLR2 (PDB ID: 5D3I) and TLR4 (PDB ID: 4G8A) were analyzed using the ClusPro 2.0 web server [58]. The resulting 3D vaccine complexes with TLR2 and TLR4 were analyzed and visualized using UCSF Chimera 1.17.1 [51]. Furthermore, PDBsum was used to identify key binding residues and analyze the interaction patterns between the vaccine and receptors [59].

## 2.9. Molecular dynamics simulation and principal component analysis

After molecular docking, the top-ranked complex was subjected to molecular dynamics simulation (MDS) to refine and analyze the interaction between the vaccine candidate and receptor proteins. This step assessed structural stability, flexibility, and compactness of the docked complexes in a dynamic environment. GROMACS 2023 was used to perform the simulations, running for 100 ns in an aqueous medium to observe molecular behavior over time [60]. The GROMOS96a force field was used to generate the required parameters for both complexes, which were then solvated in an SPC water box with a 2 Å padding to ensure proper solvation. Energy minimization was performed using the steepest descent algorithm for 100 ps to remove steric clashes. Subsequently, equilibration was conducted in two phases: first under NVT, followed by NPT, each for 100 ps at 300 K and a pressure of 1 bar. The long-range electrostatic and van der Waals forces were computed using the Particle-Mesh Ewald approach with a cutoff distance of 1 nm. Constraints were applied to maintain bond lengths and water geometry, ensuring system stability. The system temperature and pressure were controlled by applying the Berendsen thermal coupling method and the Parrinello-Rahman pressure regulation algorithm, respectively [61]. The simulation was performed under periodic boundary conditions along XYZ coordinates to prevent artifacts from interfering with the results. Post-simulation analyses were performed using the GROMACS modules [60], with structural changes and fluctuations visualized through plots and graphs generated using Xmgrace. PCA was performed using the gmx covar tool in GROMACS to examine the collective motion of the protein complexes. PCA helps identify dominant motion trends by analyzing eigenvectors and eigenvalues. The first two principal components (PC1 and PC2), which represented the most significant and independent movements, were selected for further evaluation of both simulated systems. The MM/GBSA method was applied to estimate the binding affinities using the HawkDock server [62]. This approach allowed for quantitative evaluation of the interaction strength and stability between the vaccine construct and receptor proteins, thereby supporting the outcomes of the docking and simulation studies.

## 2.10. Host-immune system simulation

The immunogenic potential of the designed vaccine construct was computationally evaluated using the C-ImmSim server [63], which simulates mammalian immune responses by integrating machine learning techniques with position-specific scoring matrices and agent-based modeling. While most parameters were kept at the default settings, modifications were made in Steps 1, 84, and 170 to optimize the analysis. Three doses were given at four-week intervals to mimic real-world vaccine administration protocols and ensure an optimal immune response according to standard vaccination guidelines. After simulation, the Simpson Index (D) was calculated to assess immune diversity and evaluate the strength of the vaccine-induced response.

## 2.11. Codon adaptation and in silico cloning

To enhance the expression of the vaccine construct in the prokaryotic host system, codon adaptation was performed by optimizing the DNA sequence to align it with the codon usage of *Escherichia coli* strain K12. The Java Codon Adaptation Tool (JCat) was used to analyze codon optimization and assess the correlation between codon usage and gene expression [64,65]. The Codon Adaptation Index (CAI) and GC content were calculated to ensure that the GC content remained

within the optimal range of 30–70% for efficient expression. *XhoI* and *BamHI* restriction sites were incorporated at the optimized sequence's 5' and 3' ends to facilitate cloning. The modified nucleotide sequence was inserted into the *E. coli* expression vector pET-28a (+) using the SnapGene restriction cloning module.

## 2.12. mRNA structure prediction

RNA secondary structure prediction for the MESV was performed using the RNAfold web server [66], which estimates the most thermodynamically stable configuration by computing base-pairing probabilities and the corresponding minimum free energy (ΔG). The prediction process is grounded in the Zuker-Stiegler dynamic programming algorithm, a widely accepted method for accurate minimum free energy (MFE)-based RNA folding simulations.

## 3. Results

### 3.1. Protein sequence retrieval

*Cryptosporidium parvum* Cp15, Cp23, and CpP2 protein sequences were obtained from the UniProt database in FASTA format. Antigenicity analysis using VaxiJen v2.0 indicated that all three proteins had scores above the 0.5 threshold, suggesting a strong potential to elicit an immune response, with Cp23 displaying the highest antigenic score (Table 1). Allergenicity evaluation *via* AllerCatPro 2.0 predicted all selected proteins to be non-allergenic. Furthermore, BLASTp comparison with the human proteome revealed no significant sequence similarity, minimizing the likelihood of autoimmune reactions. Using TMHMM 2.0, the analysis showed that absence of transmembrane helices in all prioritized proteins (S1 Fig). Collectively, these characteristics supported the selection of these proteins as suitable candidates for subsequent MESV development.

### 3.2. Epitope prediction

B-cell receptors are vital for vaccine efficacy because they initiate antibody production upon recognizing immunogenic epitopes. This triggers B-cell differentiation into plasma cells, which produce antibodies during primary responses, and memory cells, which respond rapidly during subsequent infections [67]. B-cell epitopes were predicted using ABCpred and BepiPred 3.0 servers, with one high-scoring epitope selected from each protein. The predicted antigenic regions are shown in S2 Fig. Table 2 shows the selected epitopes along with their amino acid sequences, positions, and lengths.

HTL epitopes were predicted for each of the three selected proteins using the IEDB MHC II binding prediction tool. To support the MESV construction, the top three epitopes per protein were selected based on their low percentile scores,

**Table 1. Characterization of the three prioritized proteins based on antigenicity, non-homology, molecular weight, and number of transmembrane helices.**

| Si. No. | Protein name | UniProt ID | TM helices | Antigenicity | Allergenicity | Non-homologous |
|---------|--------------|------------|------------|--------------|---------------|----------------|
| 1 | Cp15 | Q23728 | 0 | 0.5817 | Non-allergen | ✓ |
| 2 | Cp23 | Q8ITU5 | 0 | 0.7562 | Non-allergen | ✓ |
| 3 | CpP2 | Q9U553 | 0 | 0.5207 | Non-allergen | ✓ |

**Table 2. B-cell epitopes prediction for the input *Cryptosporidium parvum* protein sequences *using the* ABCpred server.**

| Si. No | Protein name | Sequence | Start position | Score | Antigenicity | Allergenicity | Toxicity |
|--------|--------------|----------|----------------|-------|--------------|---------------|----------|
| 1 | CP15 | PVAVRTHLRNMVILPE | 51 | 0.88 | 1.02 | Non-allergen | Non-toxic |
| 2 | CP23 | QKPEEPKKSEPASNNP | 50 | 0.89 | 1.04 | Non-allergen | Non-toxic |
| 3 | CpP2 | AVSGGAAAASGAAQDS | 73 | 0.88 | 1.53 | Non-allergen | Non-toxic |

indicating robust binding affinities to MHC class II alleles. Epitopes with the lowest consensus values were prioritized as leading candidates (Table 3). Upon presentation by antigen-presenting cells, naïve CD4$^+$ T cells differentiate into Th1 or Th2 subsets, which orchestrate cellular and humoral immunity, respectively. Th1 differentiation is associated with the secretion of IFN-γ, a cytokine critical for the clearance of intracellular pathogens through macrophage activation. All identified HTL epitopes were predicted to elicit IFN-γ responses, implying their potential to drive effective cell-mediated immunity. Furthermore, using the IL4Pred tool, these epitopes were classified as IL-4 inducers, suggesting an additional capacity to enhance Th2-type humoral responses [68].

CTLs recognize antigens present on MHC class I molecules and eliminate the infected cells by releasing perforin, granulysin, and granzymes, thereby providing protection against pathogens [69]. Nine CTL epitopes were selected based on their highest prediction scores, three each from Cp15 (KPVAVRTHL, TAAFIAKLR, and SVAGVYNGK), Cp23 (APQDK-PAEA, APAAQAPPA, and NPAPISNQA), and CpP2 (EEGDLGFSLF, AAQDSAPAEK, and AVSGGAAAA) (Table 4). Additionally, all predicted epitopes, including three B-cells, three HTL, and nine CTL candidates, were assessed for potential toxicity using the ToxinPred tool. All epitopes were predicted to be non-toxic, supporting their suitability for inclusion in the MESV.

### 3.3. Subunit vaccine construction and validation

To construct the MESV, KK, AAY, and GPGPG linkers were used to link the identified B-cell (3), CTL (9), and HTL (3) epitopes, respectively. These linkers enhance structural flexibility and ensure the appropriate separation of functional

**Table 3. Selected HTL epitopes from *Cryptosporidium parvum* proteins, with predictions for toxicity, antigenicity, allergenicity, IFN-γ and IL-4 production stimulation.**

| Si. No | Allele | Epitope | Percentile rank | Antigenicity | Allergenicity | Toxicity | IFN-γ inducer | IL-4 inducer |
|---|---|---|---|---|---|---|---|---|
| 1 | HLA-DPA1*01:03/DPB1*02:01 | AGVYNGKTYVTVEIK | 0.04 | 1.08 | Non-allergen | Non-toxic | Positive | Positive |
| 2 | HLA-DRB1*11:01 | KAQLAKAVKNPAPIS | 0.60 | 0.61 | Non-allergen | Non-toxic | Positive | Positive |
| 3 | HLA-DRB3*02:02 | DVLISNMSGKLSHEV | 2 | 1/02 | Non-allergen | Non-toxic | Positive | Positive |

**Table 4. Prediction of CTL epitopes from input *Cryptosporidium parvum* protein sequences using NetCTL-1.2, alongside antigenicity, allergenicity, and toxicity assessments.**

| Protein name | Epitope | HLA class I supertypes | Antigenicity | Allergenicity | Toxicity |
|---|---|---|---|---|---|
| CP15 | KPVAVRTHL | HLA-B*07:02, HLA-B*08:01, HLA-B*53:01, HLA-B*51:01, HLA-B*35:01 | 1.02 | Non-allergen | Non-toxic |
| | TAAFIAKLR | HLA-A*68:01, HLA-A*33:01, HLA-A*11:01, HLA-A*26:01, HLA-A*68:02 | 0.66 | Non-allergen | Non-toxic |
| | SVAGVYNGK | HLA-A*68:01, HLA-A*11:01, HLA-A*03:01, HLA-A*30:01, HLA-A*31:01 | 0.87 | Non-allergen | Non-toxic |
| CP23 | APQDKPAEA | HLA-B*07:02, HLA-B*08:01, HLA-B*51:01, HLA-B*35:01, HLA-B*53:01 | 0.86 | Non-allergen | Non-toxic |
| | APAAQAPPA | HLA-B*07:02, HLA-B*35:01, HLA-B*51:01, HLA-B*08:01, HLA-B*53:01 | 0.81 | Non-allergen | Non-toxic |
| | NPAPISNQA | HLA-B*35:01, HLA-B*07:02, HLA-A*68:02, HLA-B*53:01, HLA-B*08:01 | 0.57 | Non-allergen | Non-toxic |
| CpP2 | EEGDLGFSLF | HLA-B*44:03, HLA-B*44:02, HLA-B*40:01, HLA-A*01:01, HLA-B*15:01 | 0.92 | Non-allergen | Non-toxic |
| | AAQDSAPAEK | HLA-A*11:01, HLA-A*30:01, HLA-A*03:01, HLA-A*68:01, HLA-A*01:01 | 0.74 | Non-allergen | Non-toxic |
| | AVSGGAAAA | HLA-A*02:03, HLA-A*02:06, HLA-A*68:02, HLA-A*30:01, HLA-B*07:02 | 1.21 | Non-allergen | Non-toxic |

domains for proper folding and immune processing. An EAAK linker was used to incorporate the adjuvant lipoprotein LprA, a known TLR2 agonist, at the N-terminus of the construct with the aim of enhancing the immunogenicity of the vaccine. A 6×His-tag was appended to the C-terminus to facilitate downstream purification *via* affinity chromatography. The final vaccine construct comprised 450 amino acids, including the adjuvant, epitope, linker, and His-tag (Fig 2A).

The physicochemical properties were computed using the ProtParam server. The estimated molecular weight of the vaccine was 46.94 kDa, and the estimated half-life was 30h, >20h, and> 10h in mammalian reticulocytes (*in vitro*), yeast (*in vivo*), and *E. coli* (*in vivo*), respectively. The produced vaccine had a theoretical pI of 9.42, indicating the basic nature of the protein. The high aliphatic index of 76.73 and instability index of 39.71 of the vaccine verified its thermal and

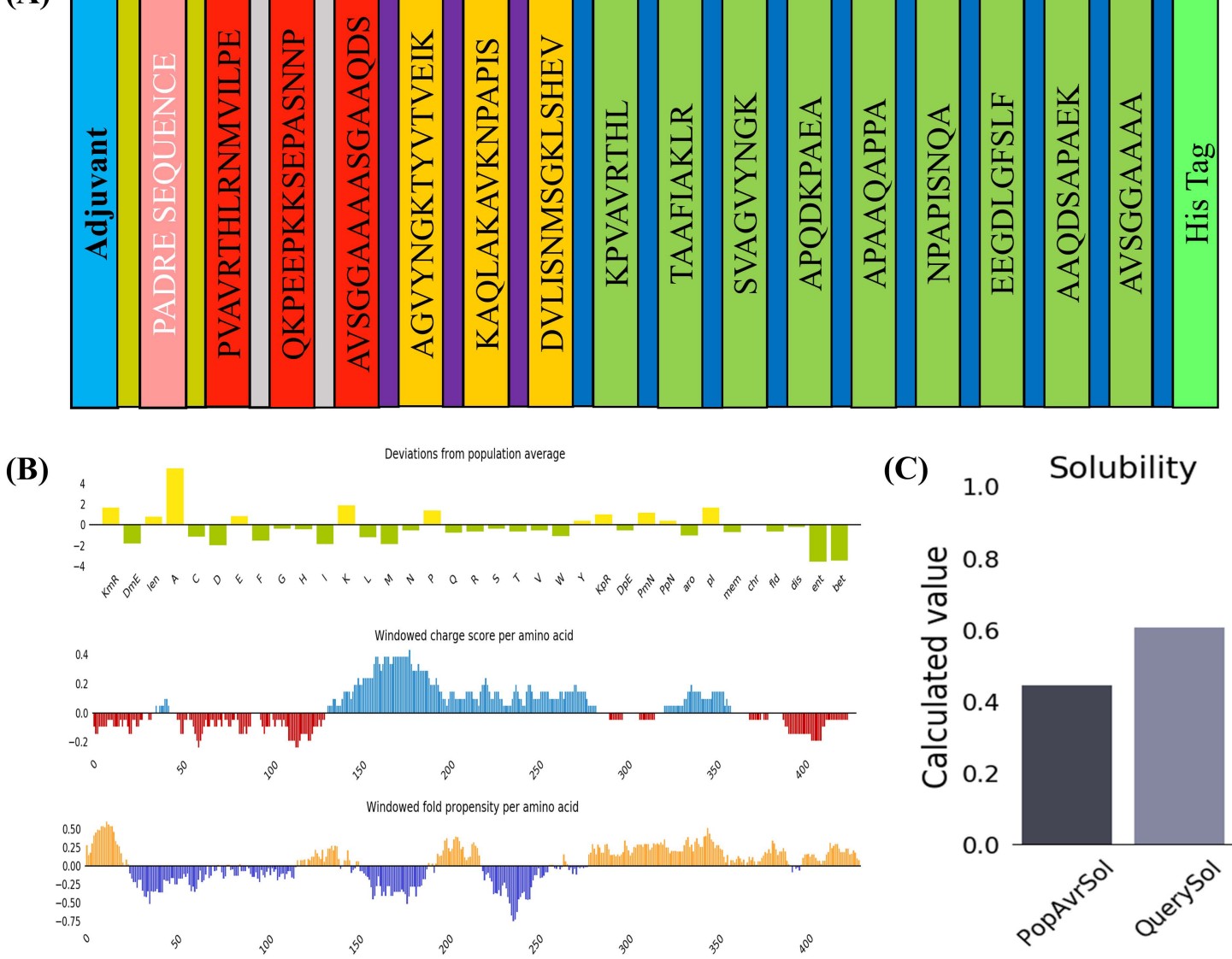

**Fig 2. Design and solubility assessment of the vaccine construct. (A)** Schematic illustration showing the key elements incorporated into the final vaccine sequence. **(B)** Graph displaying variation from population norms across 35 parameters, including regional charge and amino acid folding tendencies. **(C)** Solubility analysis of the vaccine candidate using the Protein-Sol server, benchmarked against average values from reference datasets.

conformational stability, respectively. The calculated GRAVY value of −0.106 indicates that the produced vaccine is hydrophilic (Table 5). Additionally, solubility scores of 0.512 and 0.804 were obtained for the vaccine construct using Protein-Sol and SOLpro servers, respectively (Fig 2B, 2C).

### 3.4. Structure, CABS-flex, and discontinuous antibody prediction

Secondary structure analysis using the GOR IV server revealed that the vaccine construct was comprised of 64.67% alpha-helices (291 residues), 32% random coils (144 residues), and 3.33% extended strands (15 residues). Graphical representations of the predicted secondary structures generated by both the PSIPRED and GOR IV servers are shown in Fig 3A, 3B. The initial 3D structure of the MESV construct was predicted using the trRosetta server (Fig 3C). Structural refinement was performed to improve model quality, and validation was conducted using Ramachandran plot analysis. Fig 3D and 3E illustrate the structural validation results before and after refinement, respectively. Post-refinement analysis showed that 93.8% of the amino acid residues were positioned within the most favorable regions of the Ramachandran plot, indicating a reliable and stereochemically stable model. Additionally, the ProSA-web tool assessed the overall model quality, with the Z-score improving from −8.83 in the initial structure to −7.05 after refinement, suggesting enhanced structural accuracy (Fig 3F, 3G).

CABS-flex and AGGRESCAN analyses were performed to assess the flexibility and aggregation-prone regions. The CABS-flex simulation generated ten structural models, highlighting the dynamic behavior of the vaccine protein (S3A Fig). AGGRESCAN analysis identified aggregation-prone regions within the construct, where residues with positive scores were considered prone to aggregation, whereas those with scores below zero were predicted to be soluble (S3B Fig). Among the CABS-flex models, residue 315 exhibited the highest root mean square fluctuation (RMSF) of 4.762 Å, indicating high flexibility. In contrast, residue 77 had the lowest RMSF of 1.0140 Å, reflecting structural stability in that region (S3C Fig). Furthermore, B-cell epitope prediction using the ElliPro server revealed eight linear and nine conformational epitopes, indicating surface-exposed regions of the MESV construct with the potential for antibody recognition (Fig 4 and S3 and S4 Tables).

### 3.5. Global population coverage and epitope conservation

Due to the high degree of polymorphism of MHC molecules, the distribution of HLA alleles varies across ethnic groups. This diversity limits the proportion of individuals in a population that can respond to specific T cell epitopes. Consequently,

Table 5. The physicochemical properties of the vaccine construct are predicted using the ExPASy Protparam tool.

| Physicochemical properties of vaccine | Values | Comment |
|---|---|---|
| Number of amino acids | 450 | Appropriate |
| Molecular weight | 46.94 kDa | Appropriate |
| Theoretical pI | 9.42 | Basic |
| Total number of negatively charged residues (Asp+Glu) | 49 | – |
| Total number of positively charged residues (Arg+Lys) | 62 | – |
| Total number of atoms | 6629 | – |
| Instability index | 39.71 | Stable |
| Aliphatic Index | 76.73 | Thermostable |
| Grand Average of Hydropathicity (GRAVY) | −0.106 | Hydrophilic |
| Antigenicity (VaxiJen) | 0.6617 | Antigenic |
| Antigenicity (ANTIGENpro) | 0.8551 | Antigenic |
| Allergenicity (AllerTOP) | Non-allergen | Non-allergenic |
| Solubility (Protein_Sol) | 0.607 | Soluble |
| Solubility (SOLPro) | 0.8049 | Soluble |

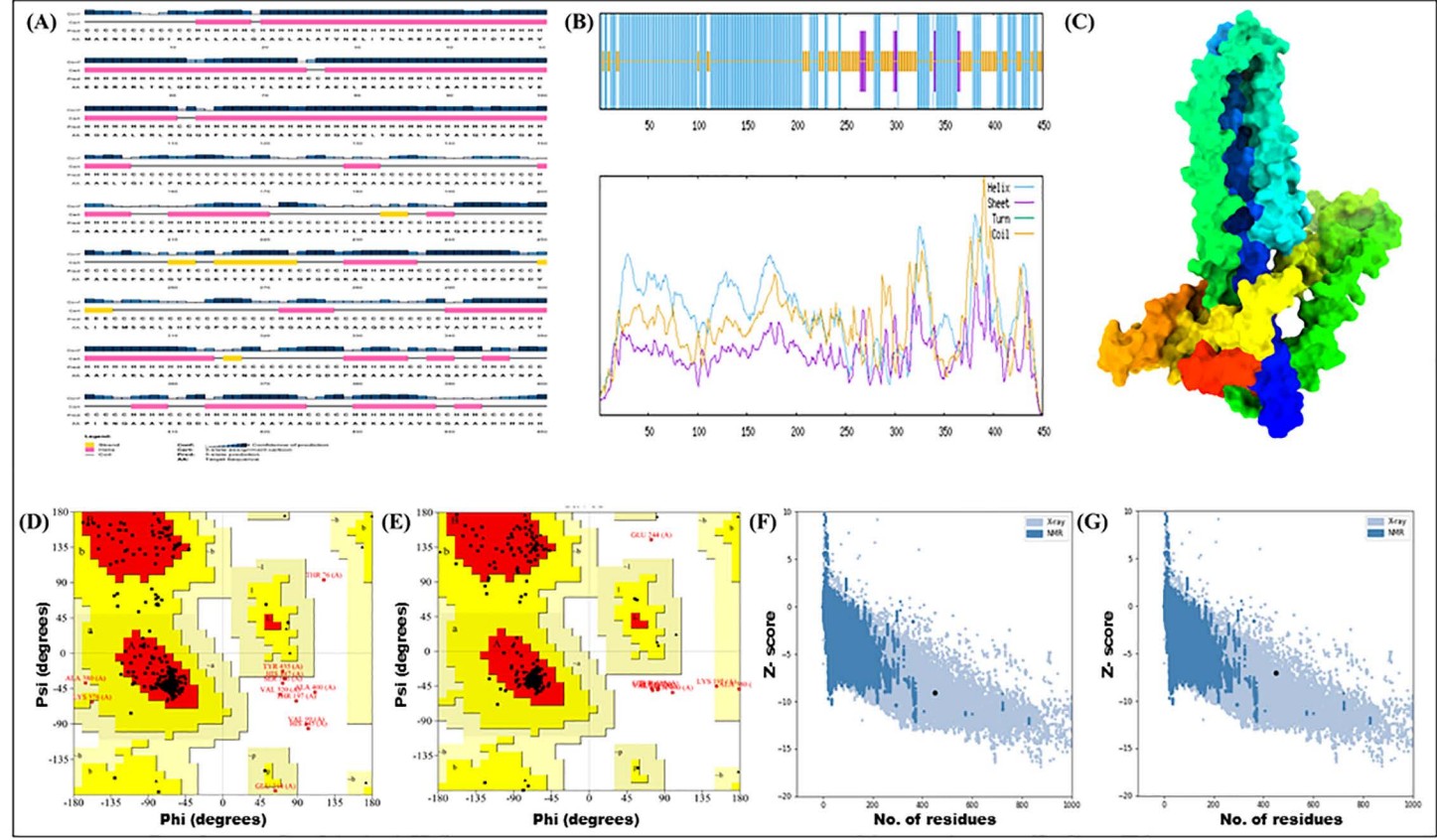

**Fig 3. Predicted structure and validation of the vaccine construct. (A)** Secondary structure prediction generated using the PSIPRED server. **(B)** Secondary structure prediction from the GOR IV server. **(C)** Predicted three-dimensional structure of the multi-epitope subunit vaccine. **(D)** and **(E)** Ramachandran plots illustrating the distribution of amino acid residues in favored, allowed, and disallowed regions before and after refinement. **(F)** and **(G)** Z-score analysis of vaccine models before and after refinement.

a peptide that acts as an epitope in one population may not be effective in another. To address this, the study aimed to identify T cell epitopes capable of binding to multiple HLA supertypes to ensure broader population coverage. The population coverage analysis, summarized in S5 Table, included 16 global regions. Europe had the highest coverage (98.91%), whereas Central Africa had the lowest coverage (70.09%) (Fig 5).

Conservation analysis was performed on epitopes from CP15, CP23, and CpP2 proteins using MEGA v7.0. In CP15, all epitopes (B-cell, HTL, CTL2, and CTL3) were fully conserved, except for CTL1 (KPVAVRTHL), which showed 97.22% conservation. For CP23, the B-cell, HTL, and CTL3 epitopes were 100% conserved, whereas CTL1 (APQDKPAEA) and CTL2 (APAAQAPPA) were 94.44% conserved. All CpP2 epitopes were fully conserved, except for the B-cell epitope (AVSGGAAAASGAAQDS), which showed 97.92% conservation. Sequence logos confirmed the high conservation of most epitopes across strains (S4 Fig). These findings highlight the predominance of highly conserved epitopes, supporting their selection for inclusion in the MESV construct and underscoring their potential for broad-spectrum immune-protection.

### 3.6. Docking, dynamics simulation, and PCA analysis

Molecular docking was performed to evaluate the interaction between MESV and the immune receptors TLR2 and TLR4, both of which play critical roles in recognizing *C. parvum* and initiating innate immune responses. For each receptor, 16

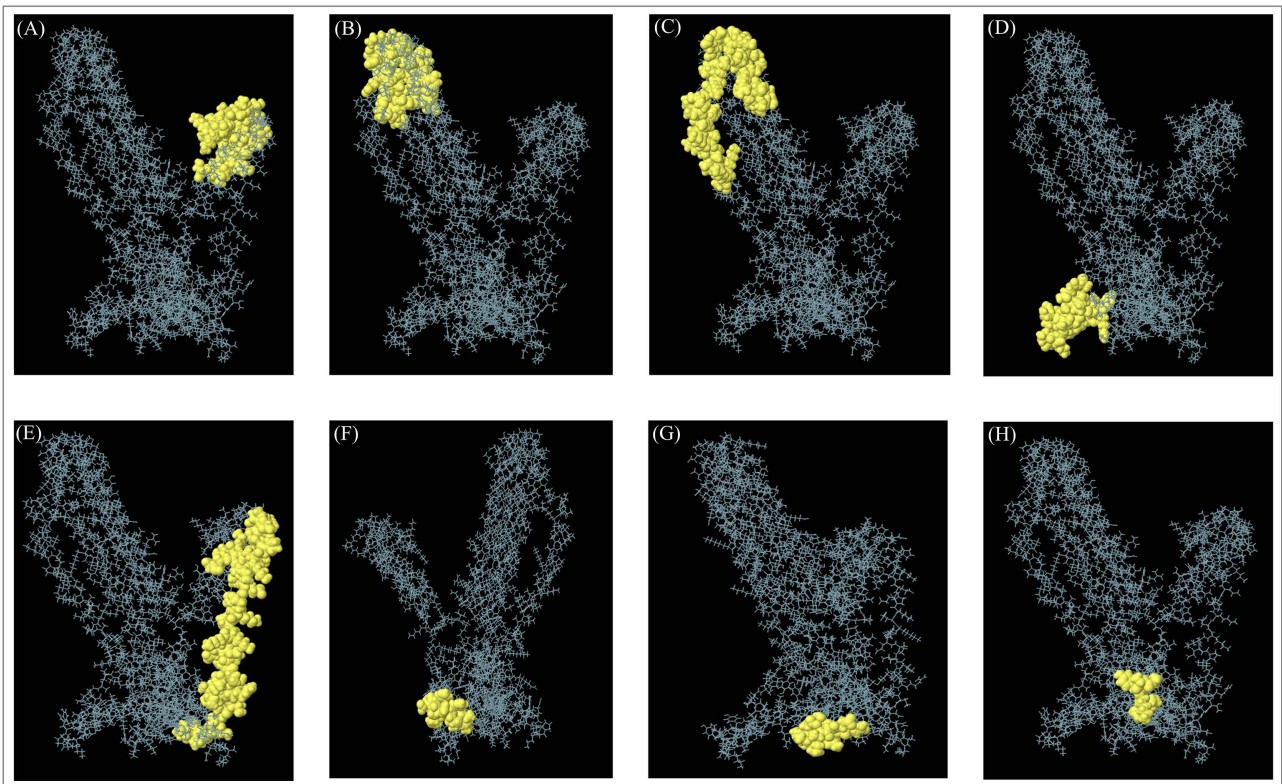

**Fig 4. Predicted conformational B-cell epitopes of the final vaccine model by the ElliPro tool.** (A–H) The ElliPro tool identified nine conformational (discontinuous) antibody epitopes highlighted in yellow within the 3D structure.

docked models were generated, and the complex with the lowest energy score was selected for further analysis. The docking results showed that the vaccine had a stronger binding affinity with TLR2 (−1328.4 kcal/mol) compared with TLR4 (−1133.3 kcal/mol), suggesting a more energetically stable interaction with TLR2. However, interaction profiling revealed that the TLR4-vaccine complex formed more stabilizing contacts, including 36 hydrogen bonds and seven salt bridges, along with 270 non-bonded interactions. In contrast, the TLR2-vaccine complex displayed 13 hydrogen bonds and 168 non-bonded contacts without any salt bridges. These findings indicate that while the TLR2 complex is energetically more favorable, the TLR4 complex may benefit from a higher number of stabilizing interactions, underscoring the vaccine's ability to effectively engage both immune receptors (Fig 6A, 6B).

To assess the structural stability and dynamic behavior of the MESV-receptor complexes, 100 ns MDS was performed for both the vaccine-TLR2 and vaccine-TLR4 systems. For the vaccine-TLR2 complex, the RMSD trajectory began with a deviation of 0.2 nm and gradually increased until 65 ns, after which it stabilized, with fluctuations remaining within a 1.5 nm range. These limited deviations indicated that the complex achieved a stable conformation during the simulation, with persistent receptor–ligand interactions (Fig 7A). The RMSF analysis revealed moderate fluctuations among the receptor residues (0.23–0.5 nm) and higher flexibility in the vaccine component (0.48–0.85 nm), suggesting dynamic regions in the antigenic structure (Fig 7B). Rg analysis showed an average value of ~4.05 nm², indicating the compactness and structural integrity of the complex over time (Fig 7C). On average, the vaccine-TLR2 complex maintained approximately 10 hydrogen bonds throughout the simulation (S5B Fig). The vaccine-TLR4 complex exhibited an initial RMSD of 0.25 nm, which rose steadily until 30 ns and stabilized thereafter at approximately 0.75 nm for the remainder of the 100 ns

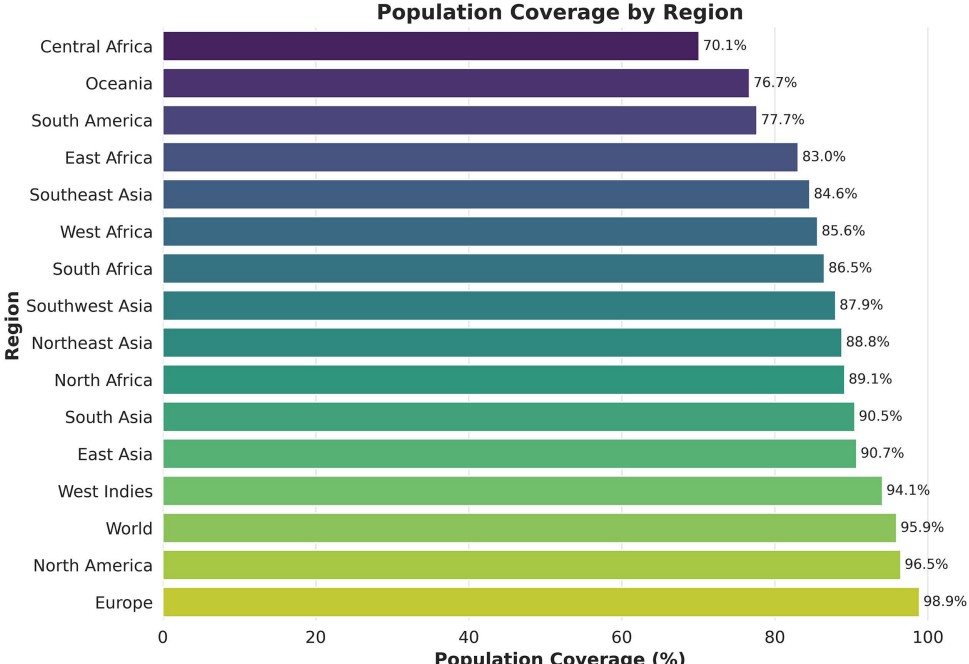

**Fig 5. Global population coverage based on the selected CTL and HTL epitopes.** The x-axis indicates the overall population coverage, incorporating both HLA class I and II alleles, while the y-axis lists the corresponding countries.

simulation, reflecting structural equilibrium (Fig 7D). RMSF revealed minor fluctuations (0.2–0.25 nm) and higher flexibility in MESV (0.55–0.8 nm) (Fig 7E). The mean Rg was ~3.40 nm², indicating a tightly packed and stable structure (Fig 7F). Similarly, ~10 hydrogen bonds were sustained on average (S5D Fig). Solvent-accessible surface area (SASA) analysis indicated an initial decrease in both complexes, suggesting hydrophilic residue burial followed by stabilization. The TLR2 complex showed an average SASA of 460 nm², whereas the TLR4 complex averaged a slightly higher SASA at 475 nm² (S5A, S5C Fig), reflecting consistent solvent interactions throughout the simulation.

PCA was performed to examine the dominant collective motion within MESV-receptor complexes. The eigenvalue (elbow) plot (Fig 8A) shows that the first two principal components capture the most significant dynamic behavior. The projection of MD trajectories onto PC1 and PC2 (Fig 8B) revealed distinct motion patterns. The TLR2-bound complex (black points) formed a compact cluster, indicating limited conformational variability, whereas the TLR4-bound complex (red points) appeared to be more dispersed, reflecting increased structural flexibility.

The binding free energy calculations using the MM/GBSA method supported these observations. The MESV-TLR2 complex exhibited a more favorable binding energy of −150.56 kcal/mol, compared with −143.39 kcal/mol for the MESV-TLR4 complex, suggesting stronger and more stable interactions with TLR2. Per-residue energy decomposition further identified key stabilizing residues contributing to binding in both complexes (Fig 8C, 8D), highlighting the molecular basis of receptor engagement and reinforcing the findings of the dynamic and energetic analyses.

### 3.7. Immune simulation

The simulation results illustrated the potent activation of both humoral and cellular immune responses following vaccine administration (Fig 9). The primary response begins with a characteristic increase in IgM antibody levels shortly after antigen exposure, indicating the activation of naïve B-cells. This was followed by a robust secondary response, marked

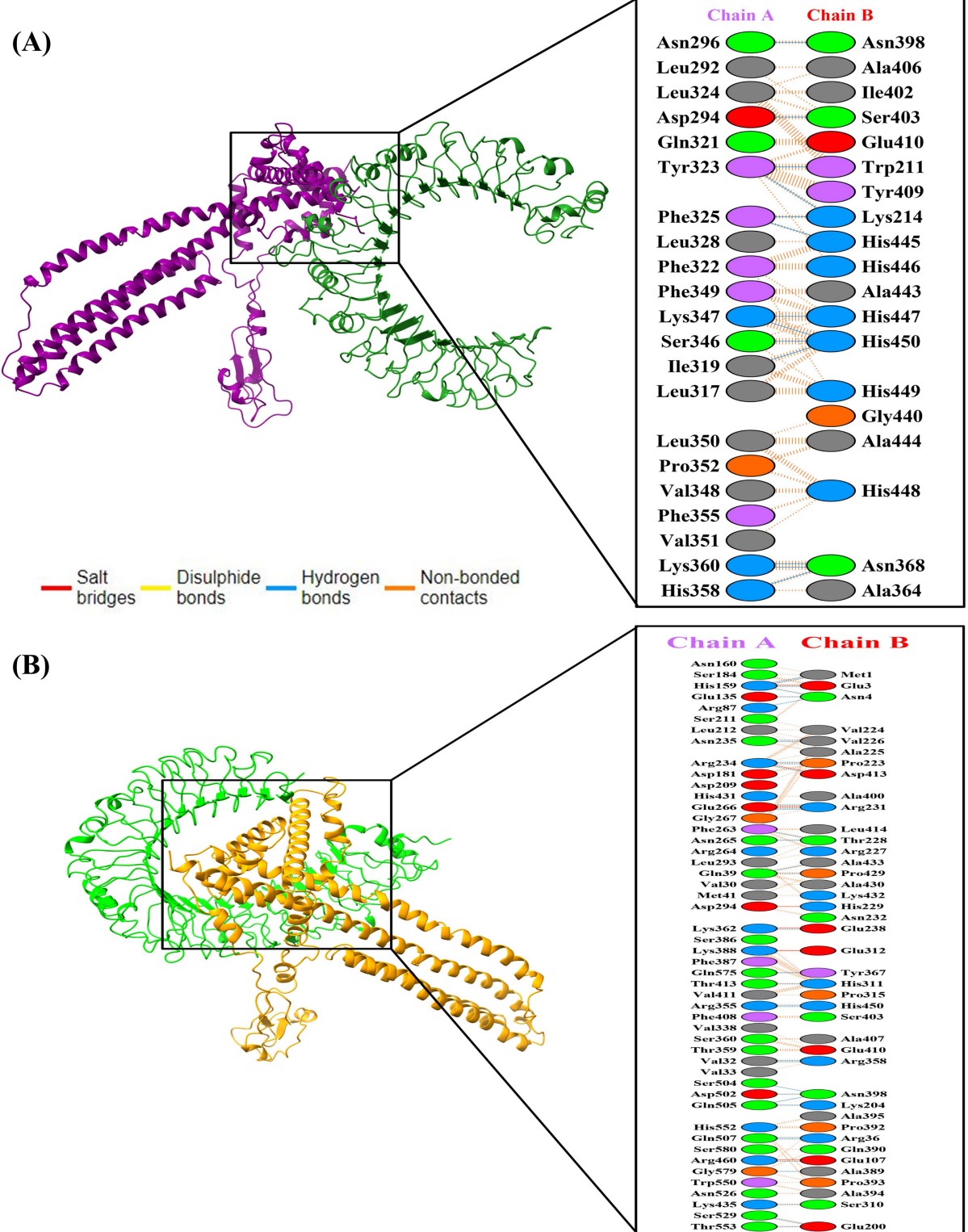

**Fig 6. Molecular docking analysis of the vaccine construct with immune receptors TLR2 and TLR4. (A)** Visualization of the docked vaccine-TLR2 complex in cartoon format, with interaction details between TLR2 (chain A) and the vaccine (chain B) generated using LigPlot. **(B)** Structural representation of the vaccine-TLR4 complex created in UCSF Chimera, highlighting the bonding interactions between TLR4 (chain A) and the vaccine construct (chain **B**).

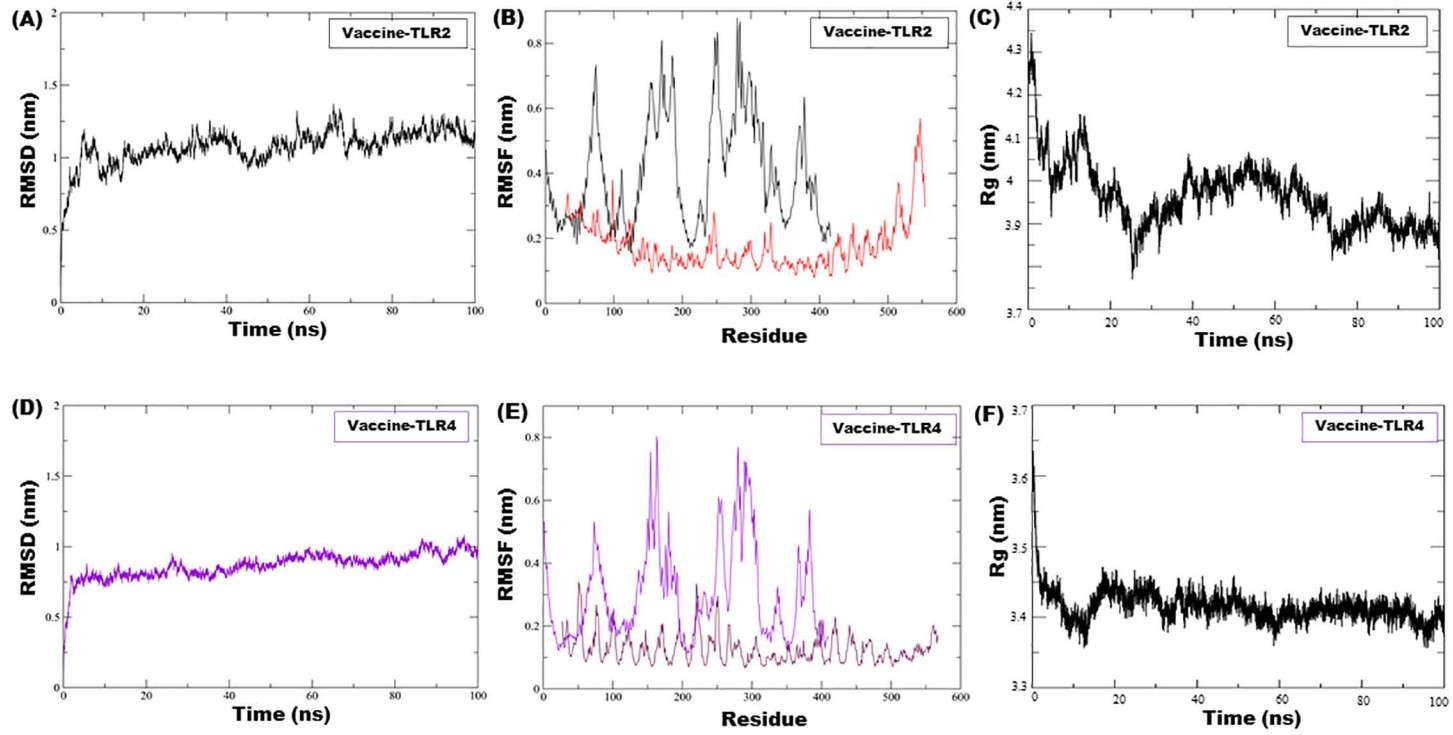

**Fig 7. Molecular dynamics simulation analysis of vaccine–receptor complexes. (A)** RMSD plot showing structural stability of the docked complex with TLR2. **(B)** RMSF plot depicting residue-level flexibility within the same complex. **(C)** The radius of gyration (Rg) plot represents the compactness of the construct and TLR2 during the simulation. **(D)** RMSD plot assessing the stability of the complex with TLR4. **(E)** RMSF analysis highlighting residue fluctuations in the TLR4-bound system. **(F)** Rg plot illustrating the structural compactness of the vaccine and TLR4 throughout the simulation.

by elevated concentrations of IgG1, IgM, and combined IgM + IgG and IgG1 + IgG2 antibodies, indicating effective class switching and memory B-cell activation (Fig 9A, 9B). Continuous stimulation leads to enhanced B-cell proliferation, a hallmark of a long-term adaptive immune response. In parallel, a notable increase in CTLs and HTLs was observed (Fig 9C, 9D), suggesting that the capability of the vaccines to stimulate cellular immunity is essential for targeting intracellular pathogens such as *C. parvum*. The vaccine also induced a strong cytokine response, especially increased secretion of IFN-γ, TGF-β, IL-2, IL-10, and IL-12, which are crucial for activating macrophages and sustaining T-cell proliferation (Fig 9E). Clonal diversity analysis based on the Simpson Index (D) reflected a broad and specific immune repertoire, confirming efficient antigen presentation and T/B-cell clonal expansion. Dendritic cells (DCs) play a pivotal role in initiating immune cascades. Following immunization, resting DC counts increased from baseline (~150 cells/mm$^3$) to approximately 200 cells/mm$^3$, alongside a moderate rise in antigen-presenting DC subsets (Fig 9F), suggesting effective antigen uptake and presentation. Collectively, these findings indicate that the designed vaccine construct elicited a strong and sustained immune response against *C. parvum*, engaging both innate and adaptive immune components essential for protective immunity.

### 3.8. In silico cloning

To ensure efficient expression in *E. coli* K12, the MESV sequence was codon optimized, yielding a GC content of 53.11% and CAI of 0.972, both indicating strong expression potential. *XhoI* and *BamHI* restriction sites were added at the 5′ and 3′ ends of the optimized sequence to facilitate directional cloning. The gene was inserted into the pET-28a (+) vector, which enabled high-level expression *via* the T7 promoter and simplifies purification through a 6 × His tag. The final recombinant

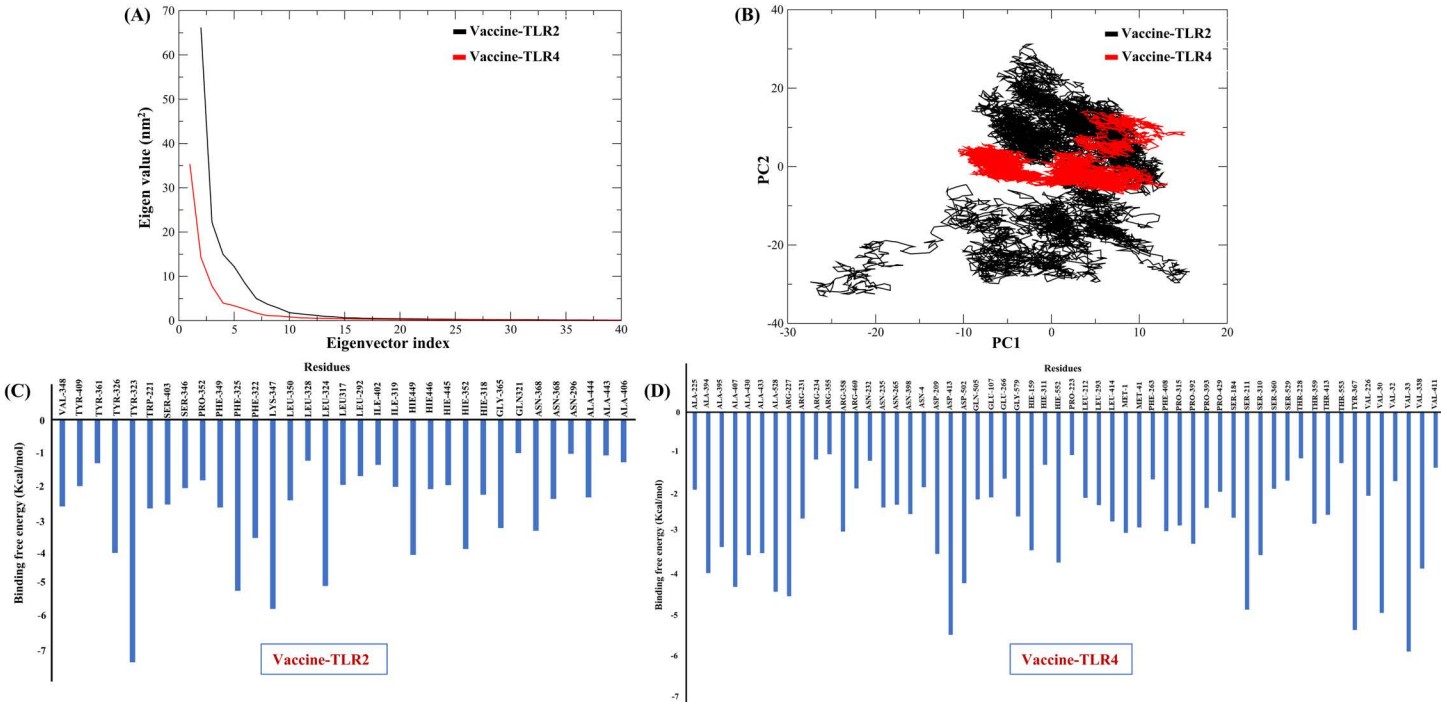

**Fig 8. Principal component analysis and binding energy decomposition of TLR2 and TLR4 complexes. (A)** Displacement of Cα atoms along the first two principal components, illustrating dominant motions in the complex systems. **(B)** A 2D projection of conformational space along eigenvectors 1 and 2 shows dynamic fluctuations for TLR2 and TLR4 interactions. **(C)** MM/GBSA per-residue energy decomposition for the TLR2 complex, identifying key contributors to binding affinity. **(D)** Binding energy profile for the TLR4 complex, highlighting residue-level contributions. Key interacting residues are marked in blue.

plasmid measured 6,685 bp, including a 1,350 bp MESV insert. SnapGene simulations confirmed successful gene integration and plasmid integrity (Fig 10).

### 3.9. MESV mRNA structure prediction

Prediction of the mRNA secondary structure based on the codon-optimized vaccine sequence yielded two representative conformations, the MFE structure and the centroid structure. The MFE structure demonstrated a highly stable configuration with a free energy value of −430.30 kcal/mol (Fig 11A), while the centroid structure exhibited a slightly higher free energy of −312.42 kcal/mol (Fig 11B). Lower energy values indicate greater thermodynamic stability, which is critical for protecting the mRNA from enzymatic degradation and maintaining its structural integrity within the host.

## 4. Discussion

Cryptosporidiosis, which is primarily caused by *C. parvum*, remains a major public health concern, particularly among immunocompromised individuals and children in developing regions. The clinical manifestations of infection include acute and chronic diarrhea, with downstream effects such as malnutrition, impaired growth, and cognitive deficits [23]. Despite decades of research, no licensed vaccine is currently available for human use, and existing pharmacological treatments, such as nitazoxanide, exhibit limited efficacy in immunocompromised hosts [22,70]. These challenges underscore the need for novel, durable, broad-spectrum prophylactic strategies. Additionally, the emergence of drug resistance and the lack of cross-protective immunity against diverse *Cryptosporidium* strains further highlight the necessity of vaccine-based

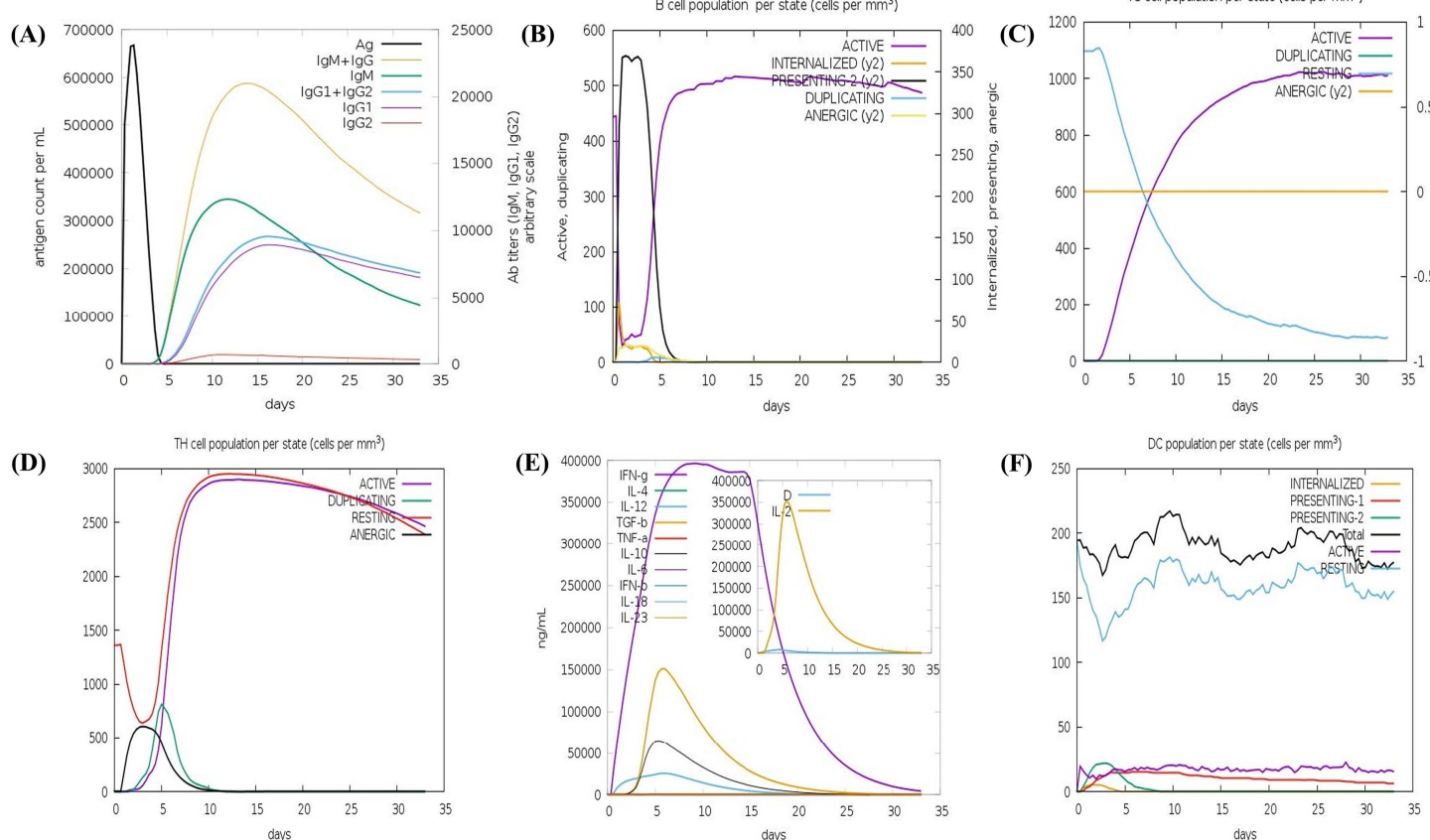

**Fig 9. Immune response simulation of the designed vaccine using the C-ImmSim server. (A)** Simulated immunoglobulin levels following antigen exposure, with distinct peaks representing various antibody isotypes. **(B)** Activation and proliferation of B-cell populations post-immunization. **(C)** Induction of cytotoxic T lymphocytes in response to the vaccine construct. **(D)** Expansion of helper T-cell populations throughout the simulation. **(E)** Cytokine release profile, highlighting IL-2 levels, accompanied by the Simpson Index [D] to assess immune diversity. **(F)** Dynamics of dendritic cell populations across different maturation states.

interventions. Immunoinformatic approaches have already been applied in MESV development against various parasitic pathogens such as *Plasmodium falciparum* [71,72], *Leishmania donovani* [73,74], *Fasciolopsis buski* [75], and *Toxoplasma gondii* [76], demonstrating promising preclinical outcomes and reinforcing the rationale for this strategy.

In this study, we designed a MESV against *C. parvum* using a comprehensive immunoinformatics pipeline. Candidate epitopes were selected based on antigenicity, immunogenicity, and conservancy, focusing on experimentally validated antigens such as Cp15, Cp23, and CpP2, which play essential roles in parasite invasion and host immune recognition [26–28]. To ensure effective antigen presentation, CTL, HTL, and B-cell epitopes were connected using chosen linkers. The AAY linker enhances CTL epitope processing through proteasomal cleavage, GPGPG promotes HTL responses while minimizing junctional immunogenicity, and KK facilitates B-cell epitope presentation and MHC-II processing [77–79]. These design strategies collectively aim to maximize proper protein folding, epitope accessibility, and overall immunogenicity. The vaccine construct was predicted to induce IFN-γ, a key cytokine mediating Th1 responses critical for controlling intracellular *C. parvum* infection, supporting both cellular and humoral immunity [80]. Additionally, lipoprotein LprA from *Mycobacterium tuberculosis* was incorporated as an adjuvant at the N-terminus using an EAAAK linker, which provides a rigid α-helical separation between the adjuvant and epitopes, improving structural stability and functional domain

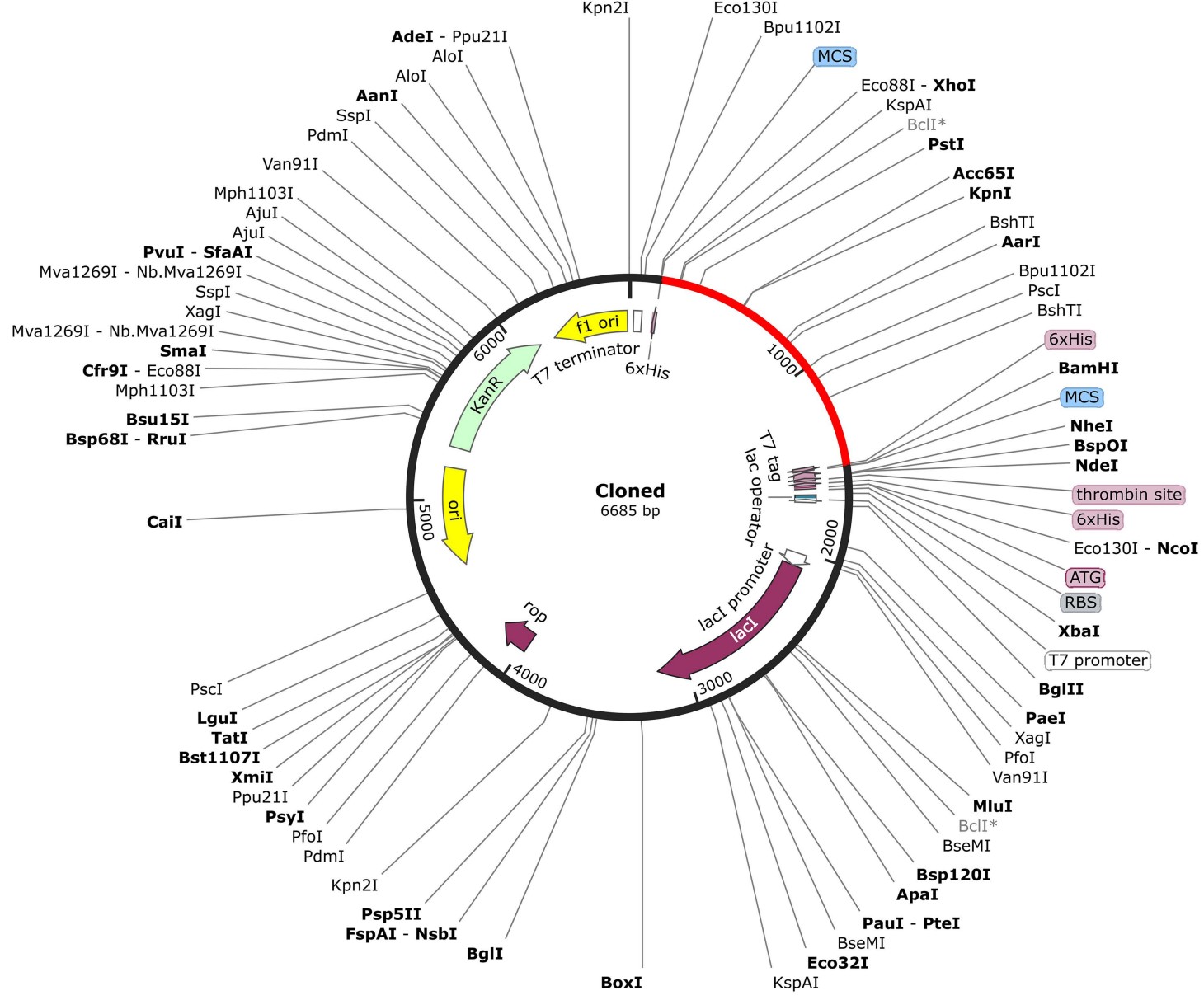

**Fig 10. *In silico* integration of the vaccine construct into the pET-28a (+) expression system.** The designed vaccine sequence (highlighted in red) was computationally inserted between the *XhoI and BamHI* restriction sites of the pET-28a (+) vector (black) to facilitate expression in a prokaryotic host.

presentation [81]. LprA enhances TLR2 signaling, dendritic cell maturation, and Th1-type immune responses [82], further promoting IFN-γ-driven control of the parasite [83]. Incorporation of these elements aimed to balance humoral and cellular immunity, a crucial factor for durable protection. Computational analyses confirmed that the MESV was antigenic, non-allergenic, non-toxic, and structurally stable. Aggrescan and CABS-flex analyses identified regions prone to aggregation, providing insights for solubility optimization [84]. Structural validation using Ramachandran plots supported stereochemical correctness, while docking studies showed strong binding affinities of MESV with TLR2 and TLR4, suggesting effective engagement of innate immune receptors [85–87]. MD simulations over 100 ns further demonstrated conformational

(A)

(B)

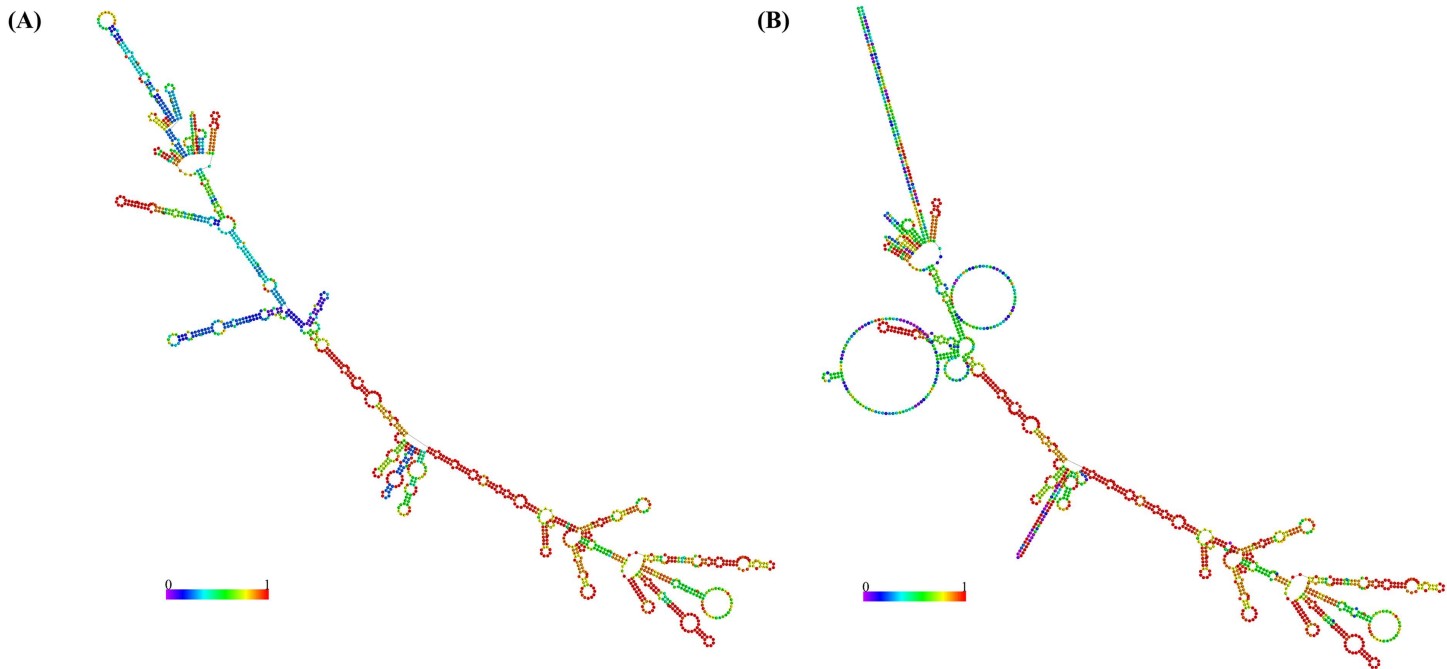

**Fig 11. mRNA secondary structure prediction of the designed vaccine using RNAfold.** (A) Optimal secondary structure corresponding to the minimum free energy. (B) Centroid structure representing the most probable base-pairing configuration.

stability, consistent with experimental findings that TLR2 and TLR4 are upregulated during *C. parvum* infection to initiate protective NF-κB–mediated immune responses [88,89]. Together, these results strengthen the likelihood of efficient receptor recognition and downstream signaling.

A key strength of this study is its methodological advancement over the MESV design reported by Dhal et al., which relied primarily on signal and hypothetical proteins and lacked rigorous structural and energetic validations. This was achieved by incorporating experimentally verified antigens and performing PCA and MM-GBSA analyses to capture dynamic stability and binding energetics. Moreover, we employed a comprehensive global population coverage analysis across 16 regions, enhancing the potential applicability of our design. The MESV construct demonstrated a potential global population coverage of 95.92%, thus highlighting its broad applicability and relevance across ethnic and geographical groups. Moreover, the prediction of the mRNA secondary structure of the MESV constructs offers an important perspective on their translational efficiency and stability, which is essential for practical vaccine development. Collectively, these analyses provide not only structural validation but also translational relevance, informing potential downstream vaccine development strategies. The *in silico* immune simulation predicted robust primary and secondary immune responses, with sustained levels of B cells, memory T cells, and elevated cytokine titers, particularly IFN-γ, TGF-β, IL-2, IL-10, and IL-12. These results suggest that the vaccine can induce both humoral and cell-mediated immunity, which is essential for protection against cryptosporidiosis. Codon optimization for expression in *E. coli* K12 demonstrated high CAI and optimal GC content, suggesting efficient transcriptional and translational expression in bacterial systems. This supports the feasibility of large-scale production of recombinant vaccines. Although experimental validation has not yet been performed, *in silico* cloning serves as a preliminary assessment of recombinant expression feasibility, facilitating downstream laboratory production and reducing trial-and-error during the early stages of vaccine development.

The present findings support the rationale for multi-epitope vaccine strategies against parasitic infections, highlighting their potential translational value. Despite these encouraging results, several limitations remain. *In silico* approaches enable rapid epitope identification, reduce dependence on pathogen cultivation, and lower early-phase development costs [90]. However, they cannot fully replicate host–pathogen complexity, predict post-translational modifications, or capture *in vivo* immunogenicity and potential off-target effects [91]. Thus, the *in silico* findings must be interpreted cautiously and complemented with *in vitro* and *in vivo* studies to confirm protective efficacy and safety. Furthermore, the antigenic diversity of *C. parvum* across different regions necessitates evaluation of cross-protection. Finally, exploring alternative adjuvants and delivery platforms may further enhance the immunogenicity and stability of the construct.

## 5. Conclusion

Cryptosporidiosis continues to pose a serious threat to global health, particularly in vulnerable populations, where treatment options remain limited. To address this, we employed a robust immunoinformatic framework to design MESV against *C. parvum* using epitopes derived from key surfaces and invasion-associated antigens (Cp15, Cp23, and CpP2). The inclusion of the TLR2-targeting adjuvant LprA was aimed at enhancing dendritic cell activation and promoting Th1-mediated responses, which are crucial for intracellular pathogen clearance. The MESV construct demonstrated favorable antigenicity, stability, and non-allergenicity, making it a viable candidate for further exploration.

Docking, MDS, and PCA confirmed stable interactions with the innate immune receptors, TLR2 and TLR4, supporting the potential of the vaccine to initiate robust receptor-mediated immune responses. Additionally, *in silico* immune simulations predicted a balanced Th1/Th2 response characterized by elevated IFN-γ and IL-4 levels. Codon optimization also supported efficient expression in *E. coli*, thereby enhancing its feasibility for large-scale production. Although these computational findings are encouraging, empirical validation using *in vitro* and *in vivo* studies is essential to confirm their immunogenicity and protective efficacy. Overall, this study provides a strong foundation for the development of an effective MESV against *C. parvum* and supports its progression toward experimental evaluation.

## Supporting information

**S1 Fig. Predicted transmembrane helices of the targeted proteins, determined using the TMHMM v2.0.** (A) Cp15, (B) Cp23, and (C) CpP2. The x-axis represents the amino acid position in the sequence, and the y-axis indicates the probability of a transmembrane helix at that position.
(JPG)

**S2 Fig. Predicted B-cell epitopes of the selected *Cryptosporidium parvum* proteins, identified using the BepiPred 3.0 server.** (A) Cp15, (B) Cp23, and (C) CpP2. The Y-axis indicates the epitope prediction score, and the X-axis represents the amino acid position within each protein. Regions above the threshold line represent predicted linear B-cell epitopes.
(TIF)

**S3 Fig.** (A) Predicted structural models generated through CABS-flex analysis. (B) AGGRESCAN output showing residues with scores greater than zero, indicating their propensity for aggregation. (C) Flexibility profile from CABS-flex, where residue fluctuations are represented as RMSF values.
(TIF)

**S4 Fig. Epitope conservation analysis.** Sequence logo visualization of predicted epitopes from (A) Cp15, (B) Cp23, and (C) CpP2 proteins. The x-axis denotes the position of amino acids within the epitope sequences, while the y-axis shows the relative frequency of each residue at those positions. The overall height of each letter stack indicates the degree of conservation, with taller stacks representing higher conservation across aligned sequences.
(TIF)

**S5 Fig. Solvent-accessible surface area (SASA) and hydrogen bond (Hb) analyses of vaccine–receptor complexes.** (A) SASA plot of the vaccine–TLR2 complex (black); (B) Hb plot of the vaccine–TLR2 complex (black); (C) SASA plot of the vaccine–TLR4 complex (black); (D) Hb plot of the vaccine–TLR4 complex (purple). (TIF)

**S1 Table. Overview of databases and web-based tools employed in the design of a multi-epitope subunit vaccine against *C. parvum.*** (DOCX)

**S2 Table. Selected HLA allele reference set of MHC-II using the IEDB server.** (DOCX)

**S3 Table. Prediction of linear (continuous) antibody epitopes using the ElliPro server.** (DOCX)

**S4 Table. ElliPro-based prediction of conformational (discontinuous) antibody epitopes.** (DOCX)

**S5 Table. Worldwide population coverage assessment of the chosen HTL and CTL epitopes.** (DOCX)

## Acknowledgments

The authors thank the Korea Institute of Toxicology (KIT) for providing essential research facilities and support.

## Author contributions

**Conceptualization:** Guneswar Sethi, Avinash Kant Lakra, Jeong Ho Hwang.

**Data curation:** Guneswar Sethi, Avinash Kant Lakra.

**Formal analysis:** Guneswar Sethi, Avinash Kant Lakra, Kirti Nirmal, Jeong Ho Hwang.

**Funding acquisition:** Jeong Ho Hwang.

**Investigation:** Jeong Ho Hwang.

**Methodology:** Guneswar Sethi, Avinash Kant Lakra, Jeong Ho Hwang.

**Resources:** Jeong Ho Hwang.

**Software:** Guneswar Sethi.

**Supervision:** Jeong Ho Hwang.

**Validation:** Guneswar Sethi, Kirti Nirmal, Jeong Ho Hwang.

**Visualization:** Guneswar Sethi.

**Writing – original draft:** Guneswar Sethi, Avinash Kant Lakra, Jeong Ho Hwang.

**Writing – review & editing:** Guneswar Sethi, Avinash Kant Lakra, Kirti Nirmal, Jeong Ho Hwang.

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
