## [Decision Letter · Decision Letter 0]

12 Aug 2025

Dear Dr. Hwang,

Thank you for submitting your manuscript to PLOS ONE. After careful consideration, we feel that it has merit but does not fully meet PLOS ONE’s publication criteria as it currently stands. Therefore, we invite you to submit a revised version of the manuscript that addresses the points raised during the review process.

We look forward to receiving your revised manuscript.

Kind regards,

Rajesh Kumar Pathak, Ph.D.

Academic Editor

PLOS ONE

Additional Editor Comments:

The manuscript has been reviewed and is found to be of interest; however, several shortcomings were identified during the review process. Please address the reviewers’ comments, expanding the introduction with current treatment strategies and their limitations, and clearly highlighting the novelty of your approach compared to similar studies. Strengthen methodological justifications, enhance the discussion with critical analysis and relevant literature, and improve figure quality, especially by removing the black background from figure 4 to enhance the overall quality and readability of the manuscript.

Reviewers' comments:

Reviewer's Responses to Questions

**Comments to the Author**

1. Is the manuscript technically sound, and do the data support the conclusions?

Reviewer #1: No

Reviewer #2: Yes

Reviewer #3: Yes

Reviewer #4: Yes

Reviewer #5: Yes

2. Has the statistical analysis been performed appropriately and rigorously?

Reviewer #1: N/A

Reviewer #2: N/A

Reviewer #3: N/A

Reviewer #4: Yes

Reviewer #5: Yes

3. Have the authors made all data underlying the findings in their manuscript fully available?

Reviewer #1: No

Reviewer #2: Yes

Reviewer #3: Yes

Reviewer #4: Yes

Reviewer #5: Yes

4. Is the manuscript presented in an intelligible fashion and written in standard English?

Reviewer #1: No

Reviewer #2: Yes

Reviewer #3: Yes

Reviewer #4: Yes

Reviewer #5: Yes

Reviewer #1: The authors present an in-silico design of a multi-epitope vaccine targeting Cryptosporidium parvum proteins Cp15, Cp23, and CpP2. While the study follows standard immunoinformatics pipelines, there are several concerns regarding the scientific novelty, rationale, and overall presentation.

Comments:

1. The target antigens should be explicitly mentioned in the abstract for clarity and completeness.

2. The Introduction section lacks a comprehensive overview of the currently available therapeutic and prophylactic strategies against C. parvum. A critical discussion of their strengths and limitations would help position the current study within the broader research context.

3. References 19–22 indicate that similar in-silico vaccine designs have already been reported. The authors should clearly articulate the novelty of their approach and justify how this work adds to the existing literature.

4. The same proteins Cp15, Cp23, and CpP2 have been previously used in related studies. Without additional experimental validation or innovative methodological contributions, the manuscript lacks sufficient novelty.

5. A dedicated discussion on the advantages and limitations of in-silico vaccine design, supported by recent literature, is essential to provide a balanced perspective.

6. The manuscript does not reference any studies that demonstrate the efficacy of multi-epitope vaccines specifically against parasitic infections. Including such references would strengthen the rationale for the approach.

7. The strain of C. parvum from which the protein sequences were retrieved is not specified. This information is critical for reproducibility and comparative evaluation.

8. The use of the LprA adjuvant (244 amino acids) in a final construct of 450 amino acids suggests a disproportionately low contribution from B-cell, CTL, and HTL epitopes. This raises concerns about the immunogenic balance of the construct and whether observed immune responses would be attributable to the adjuvant or the selected epitopes.

9. The rationale for performing in-silico cloning is unclear in the absence of downstream experimental validation. This section appears tangential and may be omitted or better justified.

10. The Discussion section largely reiterates the results without offering critical analysis or biological interpretation. It requires substantial revision to provide meaningful insights and contextual relevance.

11. Figure quality is very poor.

Overall assessment:

The manuscript, in its current form, lacks scientific rigor, novelty, and clarity in presentation. Substantial revisions are required to improve the methodological justification, contextual framing, and discussion of results. Without additional innovation or validation, the study's contribution to the field remains limited.

Reviewer #2: Major Comments

1.The author drafted the manuscript clearly.

2. I encourage the authors to provide more details about how serious the problem is. Including statistics on the mortality rate, if available, would help show the importance of this study.

3. Please provide an overview of the current treatment options available for Cryptosporidium parvum infections, including their associated side effects. Additionally, explain the significance of your study in addressing the limitations of these treatments and advancing vaccine development.

4. What is the worldwide impact or burden of this problem?

5. The presented 3.5. Global population coverage and epitope conservation explores how effectively the predicted epitopes can elicit immune responses across diverse human populations based on HLA diversity.

6. The images appear to be unclear and may require improved quality for better visibility.

7. Can the authors explain why they did not use the newer software tools available for this analysis? The tools they used are popular, but using the latest ones might add more interest.

8.I encourage the authors to approach their research from a unique perspective and clearly emphasize the novelty of their study.

9. It would be helpful to clearly highlight the key differences and novel contributions of your study in comparison to this existing work (An immunoinformatics approach for design and validation of multi-subunit vaccine against Cryptosporidium parvum) see this paper mostly like your study.

Reviewer #3: Review report

The manuscript titled “In silico design of a multi-epitope vaccine against Cryptosporidium parvum using a structural and immunoinformatics approach” presents a comprehensive computational strategy for developing a multi-epitope subunit vaccine (MESV) targeting Cryptosporidium parvum. The study integrates several advanced immunoinformatics techniques, including antigenicity prediction, molecular docking, immune simulation, and codon optimization. The work is timely, methodologically robust, and represents a valuable contribution to the field of rational vaccine design. However, minor revisions are recommended to enhance clarity and scientific rigor prior to publication.

Comments & Questions:

1. What specific thresholds were applied for antigenicity, allergenicity, and toxicity during epitope screening?

2. Why was LprA specifically selected as the TLR2 agonist over other known adjuvants?

3. Nitazoxanide is mentioned as ineffective in immunocompromised patients. Are there any other drugs under investigation that should be noted?

4. The authors mention safety concerns with live attenuated vaccines. Could you briefly comment on any documented adverse events in immunocompromised individuals?

5. Did the authors consider targeting HLA population coverage in endemic regions (e.g., Africa or Asia) during epitope selection?

6. You mention lack of transmembrane helices — was any signal peptide prediction also performed (e.g., using SignalP)? This is relevant for identifying secretory/extracellular proteins.

7. You mention PADREE. Did you mean the PADRE sequence (universal helper T-cell epitope)? Please correct spelling to avoid confusion.

8. Why was trRosetta selected over AlphaFold2 or ColabFold? These offer superior modeling and are freely accessible.

9. What criteria were used to interpret the flexibility plots? Did high flexibility regions overlap with epitope regions?

10. AGGRESCAN 3D was mentioned — were any aggregation-prone regions found? Were adjustments made accordingly?

11. If linear B-cell epitopes were already predicted in section 2.2, what was the rationale for also using conformational prediction here? Were both types used in MESV, or only one?

Reviewer #4: The manuscript entitled “In silico design of a multi-epitope vaccine against Cryptosporidium parvum using a structural and immunoinformatics approach” presents a relevant and timely contribution to the field of computational vaccine design. The study employs a variety of immunoinformatics tools to develop a multi-epitope subunit vaccine (MESV) targeting C. parvum, a pathogen of significant public health concern. The integration of antigenicity prediction, structural modeling, immune simulation, and codon optimization reflects a commendable effort toward a comprehensive in silico vaccine design pipeline.

However, in its current form, the manuscript would benefit from major revisions to improve clarity, coherence, and scientific rigor. While the methodology is broadly appropriate, several sections would benefit from clearer explanation, more detailed descriptions, and stronger justification of key design choices (e.g., antigen selection, docking protocol, and structural refinement). Additionally, the inclusion of comparative analyses or benchmarking with existing approaches would strengthen the overall impact of the study.

To enhance the reproducibility and interpretability of the findings, I recommend addressing the points outlined in the detailed review, including clarification of computational workflows, improved rationale for methodological decisions, and more thorough discussion of the biological relevance and potential limitations of the proposed vaccine construct.

Specific comments:

1. Explain how the transmission of Cryptosporidium parvum contributes to its persistence in both human and animal populations.

2. What role do thick-walled and thin-walled oocysts play in the life cycle of C. parvum, and how does this relate to disease chronicity?

3. Justify the choice of Cp15, Cp23, and CpP2 as candidate antigens in the MESV design.

4. Describe how the use of signal peptide and transmembrane helix predictions contributes to antigen selection.

5. Why was it necessary to perform allergenicity and human similarity screening for selected protein sequences? What structural refinement criteria have been selected while designing the vaccine?

6. What are the advantages of using multiple servers (e.g., ABCpred and BepiPred-3.0) for B-cell epitope prediction?

7. Explain the significance of using linkers such as AAY, GPGPG, and KK in the final MESV construct.

8. Discuss the rationale for including the PADRE sequence and a His-tag in the vaccine design.

9. What is the significance of docking in this study?

10. What is the importance of predicting both linear and conformational B-cell epitopes in vaccine design?

Reviewer #5: The manuscript entitled “In silico design of a multi-epitope vaccine against Cryptosporidium parvum

using a structural and immunoinformatics approach” offers a comprehensive computational

framework for the rational development of a multi-epitope subunit vaccine targeting C. parvum. The topic

is highly pertinent given the global health relevance of cryptosporidiosis, and the study reflects a solid

application of contemporary approaches in vaccinology and immunoinformatics. The methodology is

coherent and appropriately selected, demonstrating the potential of in silico techniques in accelerating

vaccine design.

The authors have effectively utilized various immunoinformatics tools to identify potential antigens and

epitopes, construct a chimeric vaccine sequence, and evaluate its structural and immunological properties

through docking, immune simulation, and codon optimization. The manuscript is generally clear and

flows logically from one section to the next.

A minor revision would enhance the overall clarity and impact of the work. Specifically, some

methodological descriptions could benefit from additional detail to improve reproducibility. Additionally,

a brief discussion on the potential limitations of the approach (e.g., reliance on predictive models without

experimental validation) would provide a more balanced perspective.

Comments:

1. Describe the impact of cryptosporidiosis on global health and livestock economics.

2. What challenges are associated with current treatment options for cryptosporidiosis, particularly in

immunocompromised individuals?

3. Explain the role of in silico tools in the initial screening and refinement of epitopes for vaccine design.

4. Why was LprA chosen as an adjuvant, and what is its expected immunological role in the MESV

construct?

5. What are the key physicochemical features considered essential for a successful vaccine construct, and

how were these evaluated?

6. How does the use of trRosetta and GalaxyRefine2 contribute to the structural validity of the vaccine

candidate?

7. Briefly explain the significance of the molecular docking study involving TLR2 and TLR4.

8. What is the purpose of using GROMACS and PCA in the vaccine evaluation pipeline?

**Do you want your identity to be public for this peer review?** For information about this choice, including consent withdrawal, please see our Privacy Policy

Reviewer #1: No

Reviewer #2: No

Reviewer #3: **Yes: ** RUPAL OJHA

Reviewer #4: No

Reviewer #5: No

---

## [Author Response · Author response to Decision Letter 1]

9 Sep 2025

Response to Reviewers

We sincerely thank all the reviewers for their valuable time and insightful comments. Their constructive feedback has greatly helped us to improve the quality and clarity of the manuscript. We have carefully revised the manuscript in response to each comment. The changes have been highlighted in yellow in the revised manuscript, and line numbers have been provided wherever applicable for ease of reference.

Reviewer 1

Comments:

Comment 1. The target antigens should be explicitly mentioned in the abstract for clarity and completeness.

Response: We appreciate this suggestion. We have revised the abstract to include the specific proteins used, Cp15, Cp23, and CpP2, as the target antigens (Line no. 28-29).

Comment 2. The Introduction section lacks a comprehensive overview of the currently available therapeutic and prophylactic strategies against C. parvum. A critical discussion of their strengths and limitations would help position the current study within the broader research context.

Response: We have substantially revised the introduction to include a detailed overview of current therapies (e.g., Nitazoxanide) and their limitations, as well as the lack of effective vaccines. This provides a stronger rationale for our immunoinformatics approach (line no. 78-90).

3. References 19–22 indicate that similar in-silico vaccine designs have already been reported. The authors should clearly articulate the novelty of their approach and justify how this work adds to the existing literature.

Response: Thank you for the comment. While References 19–22 report wet-lab studies involving Cp15, Cp23, and CpP2, they focus on individual or fusion protein-based vaccines and do not employ a multi-epitope subunit vaccine design. Our study is novel in integrating these experimentally validated antigens into a structure-based MESV using immunoinformatics tools, incorporating CTL, HTL, B-cell, and IFN-γ epitopes, and validating the construct through docking, immune simulation, and population coverage analysis. This approach offers a new direction for C. parvum vaccine development.

Comment 4. The same proteins Cp15, Cp23, and CpP2 have been previously used in related studies. Without additional experimental validation or innovative methodological contributions, the manuscript lacks sufficient novelty.

Response: Thank you for the comment. While Cp15, Cp23, and CpP2 have been individually validated in wet-lab studies such as Cp15-DNA vaccination in goats [1], Cp15–23 fusion protein in mice [2], and CpP2-DNA immunization in IL-12 KO mice [3], they have not been explored collectively in a multi-epitope subunit vaccine design. Our study is novel in integrating these experimentally validated antigens using a structure-based immunoinformatics approach, incorporating CTL, HTL, B-cell, and IFN-γ–inducing epitopes, and performing molecular docking, immune simulation, and population coverage analysis. This computational MESV design represents a new and rational step toward vaccine development against C. parvum.

References:

1. Sagodira, Serge, et al. "Protection of kids against Cryptosporidium parvum infection after immunization of dams with CP15-DNA." Vaccine 17.19 (1999): 2346-2355.

2. Liu, K., et al. "Divalent Cp15–23 vaccine enhances immune responses and protection against Cryptosporidium parvum infection." Parasite immunology 32.5 (2010): 335-344.

3. Benitez, Alvaro, et al. "Evaluation of DNA encoding acidic ribosomal protein P2 of Cryptosporidium parvum as a potential vaccine candidate for cryptosporidiosis." Vaccine 29.49 (2011): 9239-9245.

Comment 5. A dedicated discussion on the advantages and limitations of in-silico vaccine design, supported by recent literature, is essential to provide a balanced perspective.

Response: We thank the reviewer for this valuable suggestion. We have now included a dedicated section in the Discussion that outlines both the advantages and limitations of in-silico vaccine design, supported by recent literature (Line no. 568-577).

6. The manuscript does not reference any studies that demonstrate the efficacy of multi-epitope vaccines specifically against parasitic infections. Including such references would strengthen the rationale for the approach.

Response: Relevant references on multi-epitope vaccines for parasites such as Plasmodium falciparum, Leishmania donovani, Fasciolopsis buski, and Toxoplasma gondii have been added to the discussion to support the relevance of our strategy (Line no. 516-519).

7. The strain of C. parvum from which the protein sequences were retrieved is not specified. This information is critical for reproducibility and comparative evaluation.

Response: The protein sequences used in this study were retrieved from UniProt, with the following identifiers: Cp15 (Q23728), Cp23 (Q8ITU5), and CpP2 (Q9U553). According to UniProt, all three sequences correspond to the Cryptosporidium parvum Iowa II strain, which is commonly used in experimental studies and sequence repositories. We have now included this information in the Methods section to ensure reproducibility and facilitate comparative evaluation (Line no. 121).

8. The use of the LprA adjuvant (244 amino acids) in a final construct of 450 amino acids suggests a disproportionately low contribution from B-cell, CTL, and HTL epitopes. This raises concerns about the immunogenic balance of the construct and whether observed immune responses would be attributable to the adjuvant or the selected epitopes.

Response: We appreciate the reviewer’s concern regarding the proportion of the LprA adjuvant relative to the total length of the MESV construct. While LprA constitutes approximately half of the 450-amino-acid vaccine, its role is to enhance antigen presentation and stimulate TLR2-mediated innate immune signaling rather than to directly provide specific B-cell or T-cell epitopes. The selected CTL, HTL, and B-cell epitopes, though shorter in length, are highly immunogenic and were carefully chosen based on antigenicity, conservancy, and predicted binding affinity to human HLA molecules. Importantly, in silico immune simulations indicated robust humoral and cellular responses, including elevated levels of B cells, memory T cells, and cytokines (IFN-γ, IL-2, IL-10, IL-12), suggesting that both the adjuvant and the epitopes contribute synergistically to the overall immune response.

9. The rationale for performing in-silico cloning is unclear in the absence of downstream experimental validation. This section appears tangential and may be omitted or better justified.

Response: We thank the reviewer for raising this point. The in silico cloning and codon optimization analyses were included to evaluate the feasibility of recombinant expression in E. coli and to anticipate potential challenges in protein production. While downstream experimental validation is pending, these analyses provide a preliminary assessment of translational efficiency and guide future experimental planning. We have clarified this rationale in the Discussion section (Line no. 564-567).

10. The Discussion section largely reiterates the results without offering critical analysis or biological interpretation. It requires substantial revision to provide meaningful insights and contextual relevance.

Response: We appreciate the reviewer’s observation regarding the Discussion section. We have revised the Discussion to move beyond mere repetition of results and to provide a more critical analysis of the findings. In the revised version, we contextualize the computational outcomes within the broader biological and immunological framework of C. parvum infection, highlighting the relevance of selected epitopes, linkers, and adjuvants in eliciting both humoral and cellular immunity. We also discuss the translational implications of population coverage, mRNA secondary structure predictions, and immune simulation results, as well as the limitations of in silico approaches and the need for subsequent in vitro and in vivo validation. These revisions aim to provide meaningful insights and interpretative depth, addressing the reviewer’s concern regarding biological significance.

11. Figure quality is very poor.

Response: Thank you for your feedback. We would like to clarify that high-resolution images were uploaded during the submission process. However, due to the system's formatting, the images may appear compressed within the main PDF file. To view the figures in their original quality, kindly use the “Click here to access/download: Figure” links provided in the submission system under each figure (e.g., Figure; Fig.). We appreciate your understanding.

Reviewer 2

Major Comments

1.The author drafted the manuscript clearly.

2. I encourage the authors to provide more details about how serious the problem is. Including statistics on the mortality rate, if available, would help show the importance of this study.

Response:

We thank the reviewer for this valuable suggestion. To better emphasize the global significance of cryptosporidiosis, we have revised the introduction to include specific prevalence data in both livestock and humans. This includes infection rates ranging from 11.7% to 78% in pre-weaned calves, and human prevalence rates of up to 31.5% in low-income regions. We have also referenced the Global Enteric Multicenter Study (GEMS), which estimates nearly 202,000 child deaths annually in sub-Saharan Africa and South Asia. Additionally, we noted that C. parvum accounted for over 96% of foodborne morbidity cases between 2010 and 2020, further underlining its public health and economic impact. These additions underscore the urgency for effective vaccine development. The revised text has been highlighted in yellow from lines 68–77 in the manuscript.

3. Please provide an overview of the current treatment options available for Cryptosporidium parvum infections, including their associated side effects. Additionally, explain the significance of your study in addressing the limitations of these treatments and advancing vaccine development.

Response:

We thank the reviewer for the insightful comment. We have revised the introduction to provide an overview of current treatments for Cryptosporidium parvum. Management mainly involves supportive care and nitazoxanide, the only FDA-approved drug, which shows limited efficacy in immunocompetent individuals and is ineffective in immunocompromised patients. Other agents (paromomycin, azithromycin, clofazimine) have produced inconsistent results, and nitazoxanide is associated with gastrointestinal side effects and poor tolerability. These limitations underscore the urgent need for preventive strategies. Our study addresses this gap by designing a multi-epitope subunit vaccine targeting immunodominant C. parvum antigens to elicit robust mucosal and systemic immunity while avoiding risks linked to live or whole-parasite vaccines (Line no. 78-90).

4. What is the worldwide impact or burden of this problem?

Response:

We thank the reviewer for this insightful question. The worldwide impact and burden of cryptosporidiosis have been addressed in the manuscript. Specifically, we have included details on human prevalence, child mortality, livestock prevalence, neonatal mortality, and foodborne morbidity. These points are covered in the manuscript from line numbers 68-77.

5. The presented 3.5. Global population coverage and epitope conservation explores how effectively the predicted epitopes can elicit immune responses across diverse human populations based on HLA diversity.

Response:

We thank the reviewer for this comment. The section on global population coverage and epitope conservation was included to assess how effectively the predicted T-cell epitopes can stimulate immune responses across diverse human populations, considering the high polymorphism of HLA molecules. Our analysis ensured that selected epitopes bind to multiple HLA supertypes, maximizing coverage across 16 global regions, with Europe showing the highest coverage (98.91%) and Central Africa the lowest (70.09%) (Fig. 5; Supplementary Table 5). Conservation analysis further confirmed that most epitopes from CP15, CP23, and CpP2 proteins are highly conserved across strains, supporting their potential to induce broad-spectrum immune protection. Together, these analyses demonstrate that the predicted epitopes are likely to be effective in eliciting immune responses across diverse populations.

6. The images appear to be unclear and may require improved quality for better visibility.

Response: Thank you for your feedback. We would like to clarify that high-resolution images were uploaded during the submission process. However, due to the system's formatting, the images may appear compressed within the main PDF file. To view the figures in their original quality, we kindly suggest downloading them using the “Click here to access/download: Figure” links available at the top right-hand corner of the PDF file. Additionally, the population coverage image has been replaced with a higher-resolution version to enhance readability. We appreciate your understanding.

7. Can the authors explain why they did not use the newer software tools available for this analysis? The tools they used are popular, but using the latest ones might add more interest.

Response:

We thank the reviewer for this valuable suggestion. We acknowledge that several newer tools have been developed for population coverage and conservancy analysis. However, in our study, we used the IEDB population coverage tool, MEGA v7.0, and WebLogo v3.7.9 because they are widely accepted, validated, and extensively cited in the literature, ensuring reproducibility and comparability with previous studies. Importantly, the IEDB population coverage tool remains the standard and most reliable resource for HLA-based coverage analysis, as it integrates curated data directly from IEDB. Similarly, MEGA and WebLogo have been consistently used for sequence alignment and conservation analysis, offering robust and transparent results.

We agree that newer platforms may offer additional features; however, our selection was based on reliability, accessibility, and acceptance in immunoinformatics studies.

8. I encourage the authors to approach their research from a unique perspective and clearly emphasize the novelty of their study.

Response:

We thank the reviewer for this suggestion. In the revised manuscript, we have emphasized the novelty of our study by highlighting several key aspects: the use of experimentally validated C. parvum antigens rather than hypothetical proteins, the integration of PCA and MM-GBSA analyses to evaluate dynamic stability and binding energetics, the assessment of global population coverage, which was predicted to reach 95.92% across 16 regions, and the evaluation of mRNA secondary structure to infer translational efficiency. These elements collectively distinguish our work from previous MESV designs and provide both structural and translational insights, underscoring the unique contribution of our study to the field of cryptosporidiosis vaccine development.

9. It would be helpful to clearly highlight the key differences and novel contributions of your study in comparison to this existing work (An immunoinformatics approach for design and validation of multi-subunit vaccine against Cryptosporidium parvum) see this paper mostly like your study.

Response:

We appreciate the reviewer’s suggestion to clarify the novel contributions of our study relative to the existing work. While the referenced study employed an immunoinformatics approach to design a multi-subunit vaccine against C. parvum, our study differs in several important aspects:

Use of experimentally validated antigens: Unlike the previous work, which relied largely on predicted or hypothetical proteins, we focused on experimentally verified antigens (Cp15, Cp23, CpP2) that play established roles in parasite invasion and host immune recognition.

Rigorous structural and energetic validation: We incorporated PCA and MM-GBSA analyses to evaluate the dynamic stability and binding energetics of the vaccine construct, providing a deeper mechanistic understanding of epitope-receptor interactions.

Global population coverage analysis: Our MESV was assessed for pot

---

## [Decision Letter · Decision Letter 1]

1 Oct 2025

In silico design of a multi-epitope vaccine against Cryptosporidium parvum using structural and immunoinformatics approaches

PONE-D-25-36025R1

Dear Dr. Hwang,

We’re pleased to inform you that your manuscript has been judged scientifically suitable for publication and will be formally accepted for publication once it meets all outstanding technical requirements.

Kind regards,

Rajesh Kumar Pathak, Ph.D.

Academic Editor

PLOS ONE

Additional Editor Comments (optional):

The manuscript can be accepted for publication.

Reviewers' comments:

Reviewer's Responses to Questions

**Comments to the Author**

Reviewer #3: All comments have been addressed

Reviewer #4: All comments have been addressed

Reviewer #5: All comments have been addressed

2. Is the manuscript technically sound, and do the data support the conclusions?

Reviewer #3: Yes

Reviewer #4: Yes

Reviewer #5: Yes

3. Has the statistical analysis been performed appropriately and rigorously?

Reviewer #3: N/A

Reviewer #4: Yes

Reviewer #5: Yes

4. Have the authors made all data underlying the findings in their manuscript fully available?

Reviewer #3: Yes

Reviewer #4: Yes

Reviewer #5: Yes

5. Is the manuscript presented in an intelligible fashion and written in standard English?

Reviewer #3: Yes

Reviewer #4: Yes

Reviewer #5: Yes

Reviewer #3: After carefully reviewing the revised manuscript titled “In silico design of a multi-epitope vaccine against Cryptosporidium parvum using structural and immunoinformatics approaches,” I find that the authors have effectively addressed the concerns raised in the initial review. The resubmission reflects significant improvements in both the scientific depth and the overall quality of presentation.

The authors response to reviewer comments are comprehensive and well-justified. Methodological issues have been resolved through appropriate revisions, while interpretative ambiguities have been clarified with additional analyses and supporting evidence. The manuscript now conveys a clear and logical narrative, supported by refined technical details and strengthened validation.

Major enhancements in this version include more robust computational protocols, rigorous analysis, and stronger integration of structural biology with immunological predictions. The introduction and discussion have been notably enriched, providing greater contextual depth and highlighting the translational relevance of the findings.

Overall, the manuscript now meets the required standards of scientific rigor, methodological reliability, and clarity. The vaccine design strategy is well-executed, and the conclusions are convincingly supported by the data.

I recommend this manuscript for publication in its current form.

Reviewer #4: (No Response)

Reviewer #5: 1. The manuscript has been substantially improved in its revised form.

2. The organization and flow of the content are now much clearer.

3. The authors have provided stronger and more detailed methodological descriptions, enhancing reproducibility.

4. The discussion section is more balanced and aligns well with the results presented.

5. The responses to the reviewer comments were thorough, addressing the concerns from the initial review round.

6. The revisions have effectively resolved all previously raised issues.

7. Overall, the study now meets the standards for publication, and I recommend acceptance.

**Do you want your identity to be public for this peer review?** For information about this choice, including consent withdrawal, please see our Privacy Policy

Reviewer #3: **Yes: ** RUPAL OJHA

Reviewer #4: No

Reviewer #5: No

---

## [Editor Report · Acceptance letter]

PONE-D-25-36025R1

PLOS ONE

Dear Dr. Hwang,

I'm pleased to inform you that your manuscript has been deemed suitable for publication in PLOS ONE. Congratulations! Your manuscript is now being handed over to our production team.

Kind regards,

on behalf of

Dr. Rajesh Kumar Pathak

Academic Editor

PLOS ONE